materials science/computer modelling and simulation

alloying effects, Ni-based superalloys, atomistic simulation, dislocation nucleation, crack tip

**Author for correspondence:**
Chongyu Wang
e-mail: cywang@mail.tsinghua.edu.cn

This article has been edited by the Royal Society of Chemistry, including the commissioning, peer review process and editorial aspects up to the point of acceptance.

# Effects of Re, W and Co on dislocation nucleation at the crack tip in the γ-phase of Ni-based single-crystal superalloys by atomistic simulation

## Dianwu Wang[1], Chongyu Wang[1,2] and Tao Yu[1]

[1]Central Iron and Steel Research Institute, Beijing 100081, People's Republic of China
[2]Department of Physics, Tsinghua University, Beijing 100084, People's Republic of China

DW, 0000-0003-2471-3585; CW, 0000-0002-8147-961X; TY, 0000-0001-7131-1832

The effects of Re, W and Co on dislocation nucleation at the crack tip in Ni have been studied by the molecular dynamics method. The results show that the activation energy of dislocation nucleation is lowered by the addition of Re, W and Co; moreover, the activation energy decreases when the alloying element increases from 1 at.% to 2 at.%. The energy landscapes of the atoms are studied to elucidate these effects. Quantification analyses of the bonding strength between Ni and X (X = Re, W or Co) reveal that strong bonding between Ni and X (X = Re, W or Co) in the dislocation nucleation process can suppress the cleavage process and enhance the ability of dislocation nucleation. The surface energy and unstable stacking fault energy are also calculated to understand the alloying effects on the dislocation nucleation process. The results imply that interaction between alloying elements and Ni atoms plays a role in promoting the dislocation nucleation process at the crack tip. The ability of Re, W and Co in improving the ductility of the Ni crack system is in the order W > Re > Co. The results could provide useful information in the design of Ni-based superalloys.

## 1. Introduction

Nickel-based single-crystal (SC) superalloys have excellent mechanical properties (such as high-temperature strength, creep-resistance behaviour and low crack-growth rates) and are usually used as turbine blade materials of aerospace engines [1–4]. They

are usually alloyed with elements such as Re, W, Ta, Co, etc. Through alloying, the mechanical properties of Ni-based superalloys can be altered. Under the stress field, the material may undergo structural changes, which include both elastic and plastic deformations. The last stage of failure is often displayed with cracks in the material and the crack process is closely related with the dislocation nucleation, which is one important aspect of the ductile versus brittle response of the material. The alloying elements are reported to have influences on the dislocation nucleation process at the crack tip [5,6]. In Ni-based SC superalloys the effect of Re on the dislocation nucleation process has also been studied [6]; further studies are needed to understand the effects of different alloying elements on the dislocation nucleation process at the crack tip to gain a more complete understanding of the alloying effects in Ni-based SC superalloys from the perspective of ductile and brittle behaviour.

Dislocation emission at the crack tip is an important subject in the study of ductile versus brittle behaviour of materials [7–9]. The propagation of the crack is often the result of competition between dislocation emission and cleavage [10]. Upon dislocation emission at the crack tip, the stress around the crack tip can be relaxed and the excess energy induced by external load can be released. The result may be the crack tip blunting or the plastic deformation zone that formed around the crack tip which can shield the effect of external loadings on the crack tip [11,12]. From this aspect, the cleavage process is made more difficult by dislocation emission at the crack tip.

Several works have contributed to the related subject of dislocation nucleation at the crack tips. In studying the ductile versus brittle behaviour of the crack system, Kelly *et al.* [13] proposed the stress criterion at the crack tips, which states that when the ratio of maximum tensile stress and maximum shear stress close to the crack tip in the material is larger than the ratio of ideal cleavage stress and the ideal shear stress, then a brittle fracture may occur. If the converse is true, then the material always break with plastic deformation. While this criterion is not sufficient because the stress near the crack tip is not uniform everywhere. Rice & Thomson [8] proposed that the dislocation loop nucleation induced blunting reaction should be considered, in which they found the crystals with wide dislocation cores and small value of $\mu b/\gamma$ (where $\mu$ is the shear modulus, $b$ is the Burgers vector and $\gamma$ is the true surface energy of the crack plane) are viewed as ductile. Later works [11,14–17] analysed the dislocation nucleation from a crack tip based on the Peierls concept [18], in which the activation criterion of dislocation nucleation at the crack tip is approximately calculated with the help of the unstable stacking fault energy $\gamma_{us}$ and numerical methods for finding the transition states.

Cheng *et al.* [19] studied key factors that dominate dislocation emission from crack tip. They pointed out that existing improved nucleation models may predict higher or lower values of critical load for dislocation nucleation compared with the simulation results, and proposed the existence of competition between tension–shear coupling and nucleation–debonding coupling. Zhu *et al.* [20] studied the saddle-point curved dislocation at the crack tip and gave the activation energy for the dislocation nucleation from crack tip in Cu. The three-dimensional atomistic simulation of dislocation emission at the crack tip provides a more complete understanding of this problem. The molecular dynamics (MD) study by Hess *et al.* [21] shows that crystal orientation and temperature play important roles in the dislocation nucleation process. By performing the atomistic modelling, Gordon [5,22] showed that substitutional solute atoms in $\alpha - Fe$ have an influence on the process of dislocation nucleation at the crack tip and demonstrated that there exists a strong relationship between the solute energy landscape of the block-like shear and the activation energy for dislocation emission near the crack tip. The effect of Re on the lattice trapping effect in Ni is given by Liu *et al.* and they proposed that the dislocation emission may result from the local heterogeneities such as kinks [23]. Using atomistic simulation and the nudged elastic band method, Liu *et al.* [6] studied the effect of Re on dislocation nucleation process in Ni and found that the increase of Re concentration can lower the activation energy of dislocation nucleation. They concluded that the effect of Re on decreasing the activation energy of dislocation nucleation may relate to the local atomic structure expansion around Re atoms when dislocation passes Re atoms and the ductility of the crack in Ni can be improved by Re addition. Liu *et al.* [24] investigated the effects of Re, Co and W on the propagation of the $\gamma/\gamma'$ interface crack. However, these works only considered Re effect or lack of emphasis and comparison of alloying effects on dislocation emission from the crack tip in the $\gamma$-phase of Ni-based superalloys. Further works concerning the effects of Re, W and Co on dislocation nucleation from the crack tip in the $\gamma$-phase of Ni-based superalloys are needed.

It is experimentally observed that cracks often happen in the $\gamma$-matrix phase or near the $\gamma/\gamma'$ interface [25–27]. We study the dislocation nucleation behaviour at the crack tip in $\gamma$-phase Ni under mode I loading by atomistic simulation. The Re, W or Co atoms are randomly doped into the Ni matrix and related influences on dislocation emission at the crack tip will be studied and compared.

# 2. Models and simulation details

The MD simulation within the framework of embedded atom method (EAM) [28,29] is adopted. For the study of dislocation emission at the crack tip in systems containing Re, W or Co, we use the Ni–Al–Re [30], Ni–Al–W [31], Ni–Al–Co [32] ternary EAM potentials in our research. The Ni–Al–X (X = Re, W or Co) EAM potentials fitted the parameters obtained from first-principles calculations or from experimental results, such as lattice constants, cohesive energies, elastic constants of Ni, Al, X (X = Re, W or Co) and their compounds. The Ni–Al–Re, Ni–Al–W, Ni–Al–Co EAM potentials have been applied to predict the physical properties in the respective systems and gave reasonable results [30–32]. In particular, the Ni–Al–Re, Ni–Al–W and Ni–Al–Co EAM potentials have been applied to study the systems containing dislocations or/and cracks [6,23,24,30,33] in the Ni-based superalloys.

We adopt the $(11\bar{1})[1\bar{1}0]$ crack system in Ni under mode I loading, in which the crack lies in the $(11\bar{1})$ plane and the crack front is along the $[1\bar{1}0]$ direction. This crack system is chosen as the crack propagation and dislocations nucleation happens easily for this crack system [6,34–36]. This crack system had also been studied by Zhu et al. [20] and Liu et al. [6] and the models in current research take a similar form as their models. To study the alloying effect of Re, W or Co on the dislocation nucleation behaviour at the crack tip, 1 at.% or 2 at.% X (X = Re, W or Co) atoms are randomly doped into the Ni matrix of this crack system. The systems in current research thus are Ni matrix, Ni matrices which contain 1 at.%Re, 2 at.%Re, 1 at.%W, 2 at.%W, 1 at.%Co and 2 at.%Co, respectively.

The atomistic simulation model is cylindrical with a radius of $R = 90$ Å and the crack front is along the central (longitudinal) axis as shown in figure 1a. The crack front contains 29 unit cells in $[1\bar{1}0]$ direction (total crack front length is about 72 Å), which is sufficient to obtain an accurate activation energy of an isolated dislocation loop [6,16,20]. As the atomically sharp crack configuration in $(11\bar{1})[1\bar{1}0]$ crack system of Ni is unstable, one layer of atoms in the $(11\bar{1})$ plane behind the crack tip is removed to ensure a stable crack tip configuration. For the $(11\bar{1})[1\bar{1}0]$ crack system under mode I loading (pure tensile load applied along the $[11\bar{1}]$ axis), the activated dislocation nucleation slip plane is (1 1 1) plane, which is inclined at an angle of $\theta = 70.53°$ with respect to the $(11\bar{1})$ crack plane as shown in figure 1b. The model for the Ni system consists of 167 156 atoms. Periodic boundary condition is imposed along the $[1\bar{1}0]$ direction. The atoms within $R = 80$ Å around the cylindrical central axis (the grey atoms in figure 1a) are allowed to move and the other boundary atoms (the black atoms in figure 1a) are kept fixed during the MD relaxation. The crack systems with alloying element additions adopt the same settings as the model for the Ni system, except that the lattice parameters for the systems after doping with 1 at.% or 2 at.% X (X = Re, W or Co) are used in building the models. The lattice parameters used in the simulations are 3.520, 3.524, 3.527, 3.525, 3.530, 3.519, 3.519 for systems without alloying elements and systems with 1 at.%Re, 2 at.%Re, 1 at.%W, 2 at.%W, 1 at.%Co and 2 at.%Co, respectively. The models for calculating the lattice parameters will be introduced in the following section.

The model is incrementally loaded from $K_I = 0.60K_{Ic}$ with an increment of $0.01K_{Ic}$ until the energy barrier of dislocation nucleation is overcome and dislocation emission occurs at $K_{emit}$ ($K_{emit} = 0.86K_{Ic}$ for Ni), where $K_{Ic}$ denotes the critical stress intensity factor (theoretical Griffith stress intensity factor) and $K_{emit}$ denotes the stress intensity factor when the energy barrier of dislocation nucleation is overcome. The critical stress intensity factor characterizes the onset of the crack extension. At each loading step, the atom configurations of the model are initially determined according to linear elastic solutions in an anisotropic continuum [37,38], then the conjugate gradient method is used to relax the model while keeping the outer boundary atoms fixed.

We use the climbing image nudged elastic band (CI-NEB) method [39–42] to determine the minimum energy path (MEP) of dislocation loop emission at the crack tip. The CI-NEB method we adopted is implemented in the Large-scale Atomic/Molecular Massively Parallel Simulator (LAMMPS) [43]. The initial states for CI-NEB calculation at each $K_I$ are given by the above loading scheme. The final states at each $K_I$ for the NEB calculation are determined by unloading from $K_{emit}$ to $K_I = 0.60K_{Ic}$ with a decrement of $0.01K_{Ic}$. The boundary conditions are the same for the initial and final configurations at each specific level of stress intensity factor (SIF) $K_I$. In the CI-NEB method, system configurations (images) are inserted between the initial and final states. This method inserts system images which are interconnected by linear 'springs' in $3N$-dimensional ($N$ is the number of atoms) configurational space and this mimics an elastic band. Each configuration can be treated as an image that the spring connects. For current research 24 images (or replicas) including the initial state and the final state are used to perform the NEB calculation. The activation energy for dislocation nucleation $\Delta E_{act}$ can be obtained by subtracting the energy at the initial state from the energy at the

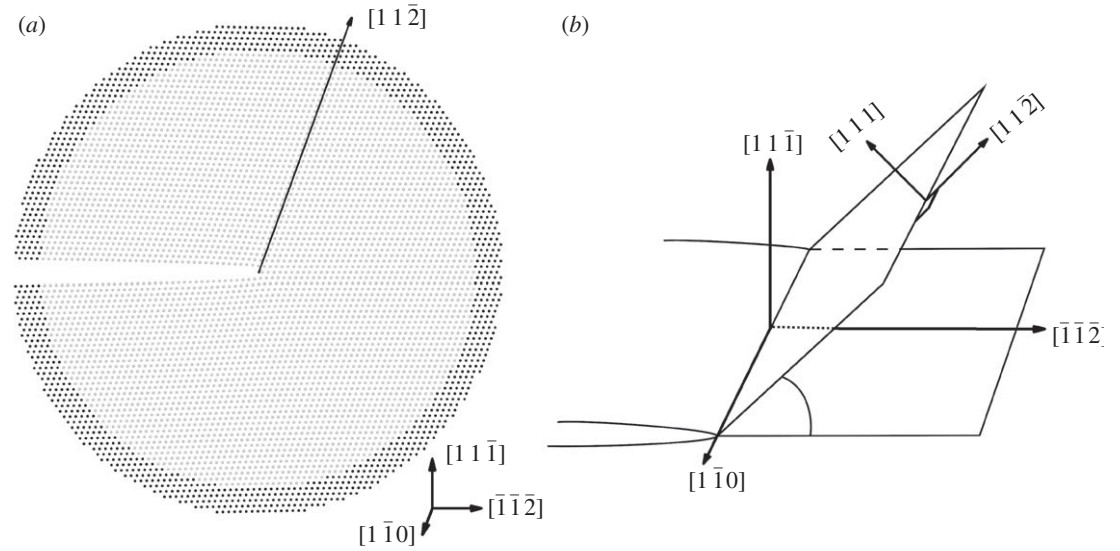

**Figure 1.** Model for dislocation loop emission at the crack tip. (*a*) Model for dislocation loop emission at the crack tip, the [1 $\bar{1}$ 0] direction is perpendicular to the paper. The grey atoms are free to move and the black boundary atoms are not allowed to move during the MD simulation. (*b*) Sketch of the inclined (1 1 1) slip plane containing the dislocation loop that is inclined at an angle $\theta = 70.53°$ with respect to the (1 1 $\bar{1}$) crack plane.

saddle-point state (saddle state) of the MEP, where the saddle-point state is the state with the highest energy in the MEP. The convergence criterion for CI-NEB calculation in current research is that the potential force on each replica vertical to the path is smaller than 0.005 eV$\mathring{A}^{-1}$.

We adopt OVITO software [44] and the common neighbour analysis (CNA) [45] algorithm implemented in OVITO to visualize atomic structures and dislocations.

# 3. Results and discussions

## 3.1. Influence of alloying elements Re, W and Co on activation energy of dislocation nucleation $\Delta E_{act}$

The minimum energy path, which is the energy variation $\Delta E$ with respect to the initial state energy, of dislocation loop emission at the loads $K_I = 0.64K_{Ic}$ and $K_I = 0.80K_{Ic}$ for Ni is shown in figure 2. The normalized reaction coordinate is used to represent the reaction coordinate. It is represented as the ratio between the hyperspace arc length along the MEP from the initial state to current state and the total hyperspace arc length along the MEP [20]. As observed from figure 2, there exists an energy barrier for dislocation nucleation in the MEP. This energy barrier characterizes the activation energy of dislocation nucleation $\Delta E_{act}$ and is a critical parameter that labels the onset of plastic deformation. As the loading increases from $K_I = 0.64K_{Ic}$ to $K_I = 0.80K_{Ic}$, the $\Delta E_{act}$ decreases from 6.32 eV to 0.53 eV.

The activation energy of dislocation nucleation ($\Delta E_{act}$) for all the systems is listed in table 1 with the loads $K_I/K_{Ic} = 0.64$, 0.68, 0.72, 0.76 and 0.80, respectively. The simulations for each concentration of randomly doped alloying element are repeated four times. The $\Delta E_{act}$ are calculated from the average value of the simulated results. Figure 3 shows the interpolated curves of the $\Delta E_{act}$ for each system, in which the cubic-spline interpolation is used. The results show that with the increase of the load $K_I$, the activation energy for dislocation nucleation $\Delta E_{act}$ decreases. All the systems with alloying element additions show a decrease of $\Delta E_{act}$ with respect to the Ni matrix in the loading range studied. The $\Delta E_{act}$ for 2 at.% X (X = Re, W or Co) addition is lower than that for 1 at.% X (X = Re, W or Co) addition (We also observe the phenomena that the $\Delta E_{act}$ of the 2 at.% Co system is larger than that of the 1 at.% Co system as seen in the electronic supplementary material, table S4. (A similar phenomena is also seen in the work of Liu *et al.* [6].) It is due to the random fluctuation of atoms that the number of Co atoms at the dislocation loop in the 2 at.% Co system is lower than that in the 1 at.% Co system. The overall comparative results of the averaged $\Delta E_{act}$ are not influenced by this fluctuation and the averaged $\Delta E_{act}$ is reasonable to reduce the influence of the random fluctuation of atoms.). Among

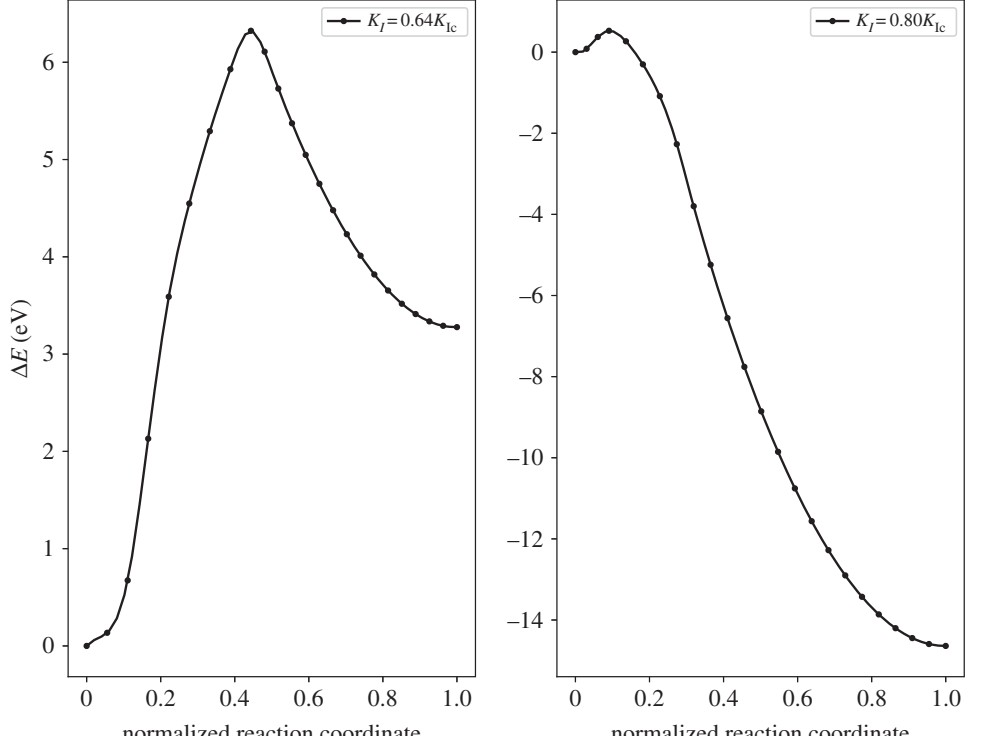

**Figure 2.** Minimum energy paths for dislocation nucleation in Ni at $K_I = 0.64K_{Ic}$ and $K_I = 0.80K_{Ic}$.

**Table 1.** Activation energy of dislocation nucleation $\Delta E_{act}$ (eV) at different loads. The standard errors for each value of $\Delta E_{act}$ are given in the parentheses.

| system | $0.64K_{Ic}$ | $0.68K_{Ic}$ | $0.72K_{Ic}$ | $0.76K_{Ic}$ | $0.80K_{Ic}$ |
|---|---|---|---|---|---|
| Ni | 6.32 | 4.15 | 2.34 | 1.22 | 0.53 |
| 1 at.%Re | 5.98 (0.09) | 3.78 (0.08) | 2.07 (0.07) | 1.04 (0.06) | 0.39 (0.03) |
| 2 at.%Re | 5.12 (0.09) | 3.02 (0.12) | 1.57 (0.07) | 0.60 (0.07) | 0.16 (0.07) |
| 1 at.%W | 5.80 (0.14) | 3.58 (0.10) | 1.93 (0.08) | 0.94 (0.08) | 0.34 (0.04) |
| 2 at.%W | 4.38 (0.17) | 2.58 (0.27) | 1.13 (0.09) | 0.33 (0.07) | 0.05 (0.05) |
| 1 at.%Co | 6.13 (0.05) | 3.98 (0.05) | 2.21 (0.04) | 1.14 (0.02) | 0.49 (0.02) |
| 2 at.%Co | 6.02 (0.09) | 3.83 (0.08) | 2.14 (0.04) | 1.11 (0.03) | 0.46 (0.02) |

the systems studied, W has the strongest ability in reducing $\Delta E_{act}$ than systems containing Re or Co when concentration changes from 1 at.% to 2 at.%, which indicates that W may be a better element in improving the ductility of the crack system in Ni compared with Re or Co.

## 3.2. Dislocation loop nucleation at the crack tip

In our study, we observed the dislocation nucleation at the crack tip, and the (1 1 1) slip plane for dislocation nucleation under mode I loading is inclined at $\theta = 70.53°$ with respect to the (1 1 $\bar{1}$) crack plane. The saddle state dislocation loop for the Ni matrix system under $K_I = 0.64K_{Ic}$ is shown in figure 4 and the face-centred cubic (FCC) structure atoms are removed for a clearer view of the atomic structure. The saddle state shows an incipient dislocation loop bowing out at the crack tip. This dislocation is a $a_0/6\langle 1\,1\,2 \rangle$ Shockley partial dislocation, where $a_0$ is the lattice constant. We observed that the Ni matrices with X (X = Re, W or Co) addition also show similar $a_0/6\langle 1\,1\,2 \rangle$ type dislocation loops.

In order to examine the saddle state configuration for dislocation nucleation, the relative shear and normal displacement of the adjacent upper and lower atomic layers across the inclined (1 1 1) slip

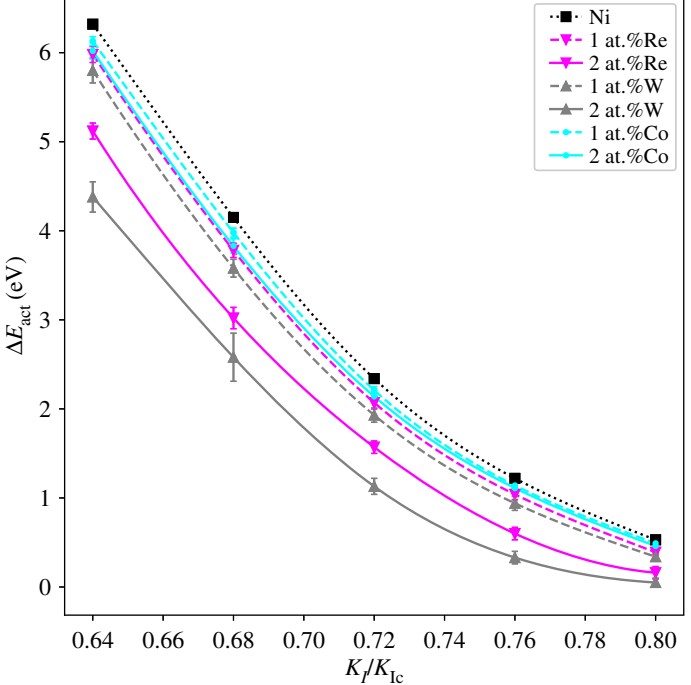

**Figure 3.** Activation energy of dislocation nucleation ($\Delta E_{act}$) at the crack tip for Ni matrix and Ni matrices with 1 at.% or 2 at.% X (X = Re, W or Co). The error bars representing the standard errors are also shown in the figure.

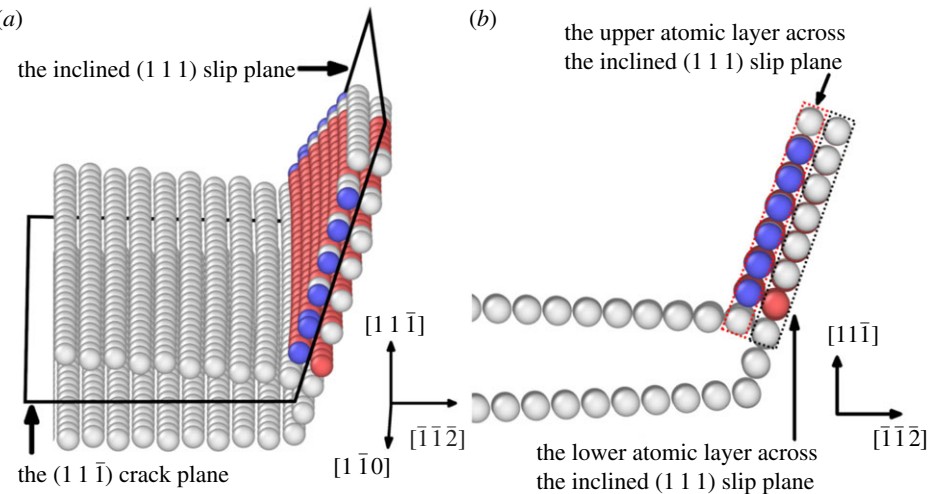

**Figure 4.** Saddle state dislocation loop emitted at the crack tip in Ni at $K_I = 0.64K_{Ic}$. The atoms belonging to the face-centred cubic (FCC) structure are removed for clarity. The red atoms belong to the hexagonal closed packed (HCP) structure and the blue atoms belong to the body-centred cubic (BCC) structure. (a) The dislocation loop configuration; (b) the upper and lower atomic layers across the inclined (1 1 1) slip plane for dislocation loop nucleation. The inclined (1 1 1) slip plane for dislocation nucleation here is defined as a mathematical cut of zero thickness between the upper and lower atomic layers.

plane (figure 4) are studied. The relative shear and normal displacements are given with respect to the initial states. Ni matrix and Ni matrices with 2 at.%X (X = Re, W or Co) addition at the load of $K_I = 0.64K_{Ic}$ are used. The relative shear displacement in [11$\bar{2}$] direction for the upper and lower atomic layers across the inclined (1 1 1) slip plane are given in figure 5. The coordinates and displacements are all normalized by the Burgers vector $b = a_0/6[11\bar{2}]$ and the continuous normalized relative shear displacement contour is obtained by cubic-spline interpolation of the normalized relative shear displacement at each atomic site. The atoms in the upper atomic layer and the lower atomic layer across the inclined (1 1 1) slip plane in figure 5 are projected onto the (1 1 1) slip plane as triangles and inverted triangles, respectively. The black triangles and inverted black triangles

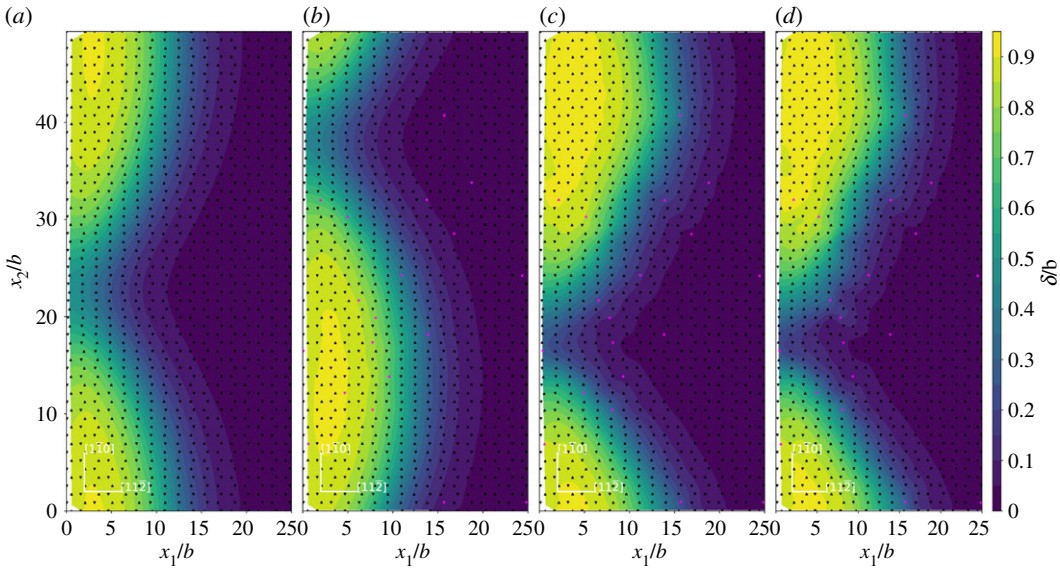

**Figure 5.** Relative shear displacement in $[1\,1\,\overline{2}]$ direction for the adjacent upper and lower atomic layers across the inclined (1 1 1) slip plane at the load of $K_I = 0.64 K_{Ic}$. The relative shear displacement is given with respect to the initial state. (*a*) Ni matrix system; (*b*) 2 at.%Co system; (*c*) 2 at.%Re system and (*d*) 2 at.%W system. The coordinates and the relative shear displacement $\delta$ are normalized by the Burgers vector *b*. The triangles and inverted triangles represent the atoms in the upper atom layer and the lower atomic layer across the inclined (1 1 1) slip plane. Black triangles and inverted black triangles represent Ni atoms, pink triangles and inverted pink triangles represent X (X = Re, W or Co) atoms.

represent the Ni atoms, and the pink triangles and inverted pink triangles represent the X (X = Re, W or Co) atoms. As shown in figure 5, the contour line with the displacement approximately around $b/2$ is the locus of the dislocation loop, which is a $a_0/6\langle 1\,1\,2\rangle$ Shockley partial dislocation. It is evident that in forming the dislocation loop, the relative shear displacement has a non-uniform distribution. This non-uniform distribution of shear displacement across the slip plane, instead of forming a uniform shear displacement distribution, can make the dislocation emission process easier. It can also be seen that the dislocation loop near the crack tip is localized in the saddle state. Thus, the fixed boundary effect on our transition state is expected to be weak and the in-plain radius R in our model is reasonable from this respect. In the area enclosed by the dislocation loop and the crack front, the maximum relative shear displacement is around $1b$. This slipped region is identified to be swept by a $a_0/6\langle 1\,1\,2\rangle$ Shockley partial dislocation. Across the inclined (1 1 1) slip plane, the area with the relative shear displacement smaller than $b/2$ is not swept by the dislocation. The projected atoms labelled as triangles and inverted triangles on the inclined (1 1 1) slip plane also show the arrangement of atoms across the slip plane in the dislocation core has a different structure from the atoms in the region swept by the dislocation and the region not swept by the dislocation. The relative shear displacement is slightly affected by alloying element additions of X (X = Re, W or Co) atoms.

Before proceeding to the following discussions, there is the need to visualize the local atomic structure of atoms across the inclined (1 1 1) slip plane. This will be helpful for understanding the local atomic structure across the inclined (1 1 1) slip plane and will be beneficial for the following discussions. The atomic structure of atoms across the inclined (1 1 1) slip plane is shown in figure 6. The alloying element X (X = Re, W or Co) in the lower atomic layer and the upper atomic layer across the inclined (1 1 1) slip plane are shown as magenta coloured atoms. The 12 nearest-neighbour atoms to the alloying element X (X = Re, W or Co) in the lower atomic layer across the inclined (1 1 1) slip plane is denoted as $L1 - L12$. Correspondingly, the 12 nearest-neighbour atoms to the alloying element X (X = Re, W or Co) in the upper atomic layer across the inclined (1 1 1) slip plane is denoted as $U1 - U12$. Furthermore, the Ni atoms in figure 6*a,b* labelled with black dots are denoted as $L3'$ and $U6'$, respectively. When the system configuration evolves from the initial state to the saddle state, the distance between $L3'$ Ni atom and the alloying atom in figure 6*a* decreases. Correspondingly, the distance between $U6'$ Ni atom and the alloying atom in figure 6*b* also decreases as the system configuration evolves from the initial to the saddle state. As shown in figure 5, for the atoms in the region already swept by the dislocation the $L3'$ atom becomes the first nearest neighbour of the

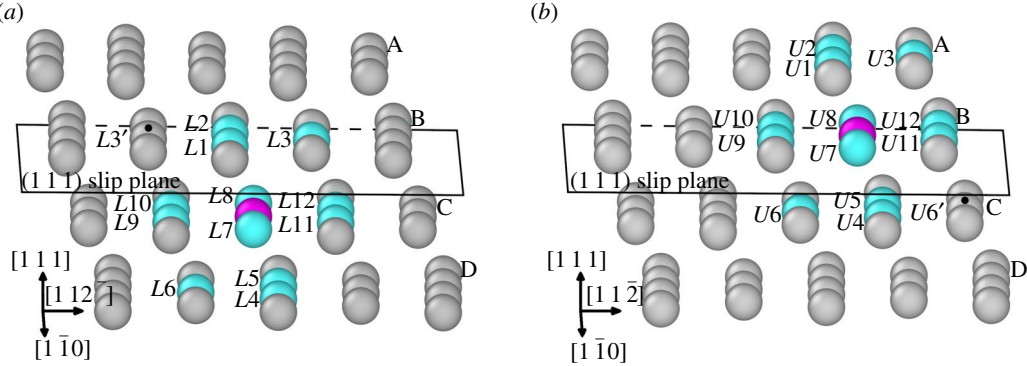

**Figure 6.** Atomic structure across the inclined (1 1 1) slip plane. Magenta atom denotes X (X = Re, W or Co) atom, grey atoms denote Ni atoms, cyan atoms labelled with $L1 - L12$ or $U1 - U12$ denote the first nearest-neighbour Ni atoms of the X (X = Re, W or Co) atom. A, B, C, D represent the atomic layers parallel to the (1 1 1) slip plane of dislocation nucleation. The atoms marked with black dots are labelled as $L3'$ and $U6'$, respectively. (a) Alloying atom X (X = Re, W or Co) resides in the lower atomic layer (layer C) across the inclined (1 1 1) slip plane. (b) Alloying atom X (X = Re, W or Co) resides in the upper atomic layer (layer B) across the inclined (1 1 1) slip plane.

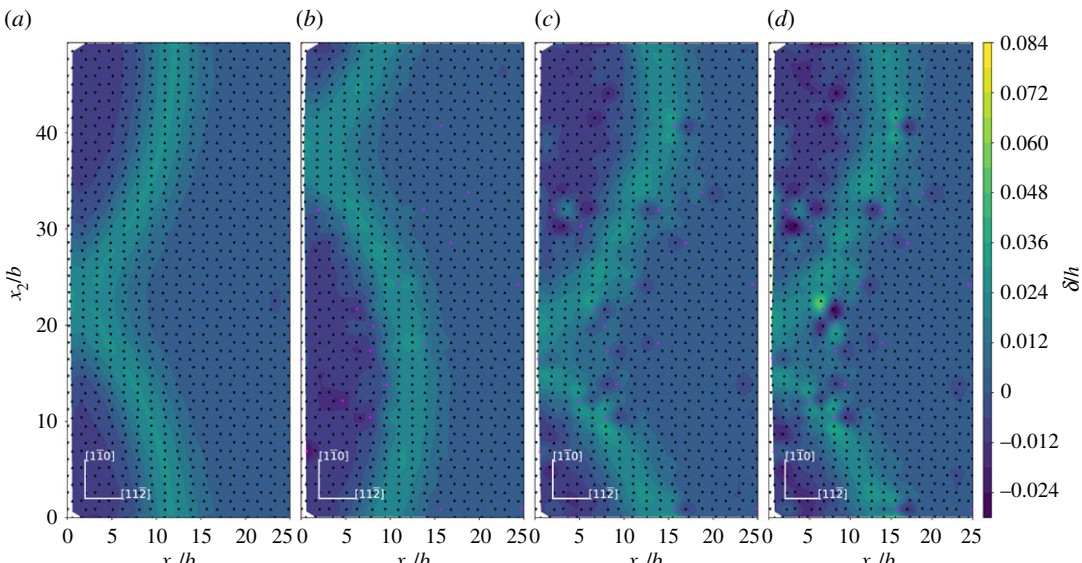

**Figure 7.** Relative normal displacement across the inclined (1 1 1) slip plane. (a) Ni matrix system; (b) 2 at.%Co system; (c) 2 at.%Re system and (d) 2 at.%W system. The coordinates and the relative normal displacements $\delta$ are normalized by the (1 1 1) interplanar spacing $h$. Triangles and inverted triangles represent the upper and lower atomic layer atoms across the inclined (1 1 1) slip plane. Black triangles and inverted black triangles represent Ni atoms, pink triangles and inverted pink triangles represent X (X = Re, W or Co) atoms.

alloying element instead of the $L3$ atom. The same situation also applies to the $U6'$ and $U6$ atom in figure 6b. When the atoms are swept by the dislocation, the $U6'$ atom becomes the first nearest-neighbour atom of the alloying element instead of the $U6$ atom. In the following discussions, if the atom is termed with $L1 - L12$, $U1 - U12$, $L3'$ or $U6'$, it refers to the neighbouring Ni atom of the alloying atom as indicated by figure 6.

The relative normal displacement of the adjacent upper and lower atomic layers across the inclined (1 1 1) slip plane is shown in figure 7 and the displacement is normalized with respect to the (1 1 1) interplanar spacing $h$. It is observed that in the saddle state the atoms at or near the dislocation core across the inclined (1 1 1) slip plane have the maximum normal displacement, which indicates a slightly expanded local structure near the dislocation core. The alloying elements studied in current paper can affect the local normal displacement of their neighbouring Ni atoms across the inclined (1 1 1) slip plane. For the neighbouring Ni atoms ($L3$, $L3'$, $U6$, $U6'$) of the alloying element across the

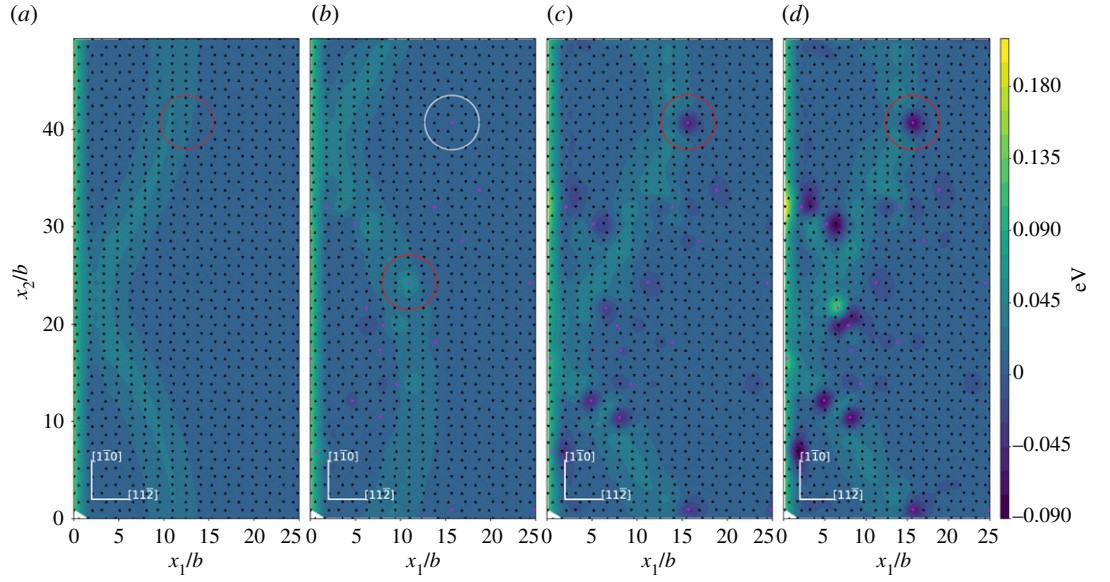

**Figure 8.** $\Delta E_j$ contour in the lower atomic layer across the inclined (1 1 1) slip plane for dislocation nucleation at the load $K_I = 0.64 K_{Ic}$. (a) Ni system; (b) 2 at.%Co system; (c) 2 at.%Re system and (d) 2 at.%W system. The triangles and inverted triangles represent the upper and lower atomic layer atoms across the inclined (1 1 1) slip plane. Black triangles and inverted black triangles represent Ni atoms, pink triangles and inverted pink triangles represent X (X = Re, W or Co) atoms.

inclined (1 1 1) slip plane, most of their relative normal displacement become smaller near the crack tip and near the dislocation loop due to Ni–X (X = Re, W or Co) interaction. Among the three alloying elements studied, Co shows relatively weak effect in affecting the relative normal displacement of its neighbouring Ni atoms (L3, L3′, U6, U6′) across the inclined (1 1 1) slip plane. The ability of Re, W and Co in affecting the relative normal displacement of their neighbouring Ni atoms across the inclined (1 1 1) slip plane are in the order W > Re > Co.

## 3.3. Reason for the effects of alloying element X (X = Re, W or Co) on decreasing activation energy of dislocation nucleation at the crack tip

The reason that alloying elements can decrease the activation energy of dislocation nucleation can be explained by the energy difference of atom $j$ between the saddle state and the initial state ($\Delta E_j$). The sum of energy difference of atom $j$ between the saddle state and the initial state gives the energy difference between the saddle state and the initial state of the system. The relations are given as follows [6]:

$$\Delta E_j = E_j^{sad} - E_j^{ini} \tag{3.1}$$

and

$$\Delta E_{act} = \sum_{j=1}^{N} \Delta E_j, \tag{3.2}$$

where equation (3.1) gives the energy difference of atom $j$ between the saddle state and the initial state, $E_j^{sad}$ and $E_j^{ini}$ denote the energy of the atom $j$ in the saddle state and the initial state, respectively. $N$ is the number of atoms in the system.

The systems for studying the $\Delta E_j$ contour in current research are the Ni matrix and the Ni matrices with 2 at.%X (X = Re, W or Co) addition at the load $K_I = 0.64 K_{Ic}$. The contours of $\Delta E_j$ for the lower and upper atomic layers across the inclined (1 1 1) slip plane are shown in figures 8 and 9, respectively. From the results, we can identify the dislocation core region from the arrangement of the projected triangle and inverted triangle atoms on the inclined (1 1 1) slip plane. The relatively higher value of $\Delta E_j$ distribution in the contours can also be used to identify the dislocation core region. In the region near the dislocation loop, the upper atomic layer atoms across the inclined (1 1 1) slip plane have relatively higher values

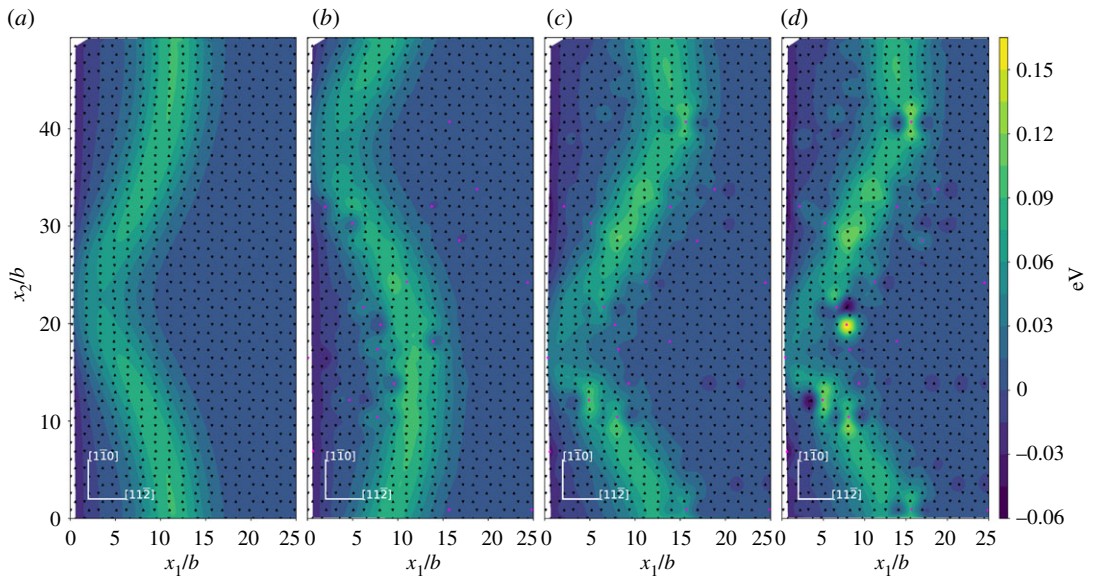

**Figure 9.** $\Delta E_j$ contour in the upper atomic layer across the inclined (1 1 1) slip plane for dislocation nucleation at the load $K_I = 0.64K_{Ic}$. (a) Ni system; (b) 2 at.%Co system; (c) 2 at.%Re system and (d) 2 at.%W system. The triangles and inverted triangles represent the upper and lower atomic layer atoms across the inclined (1 1 1) slip plane. Black triangles and inverted black triangles represent Ni atoms, pink triangles and inverted pink triangles represent X (X = Re, W or Co) atoms.

of $\Delta E_j$ than that of the lower atomic layer atoms. This indicates that near the dislocation loop the lower atomic layer atoms across the inclined (1 1 1) slip plane are generally more stable than the upper atomic layer atoms when the system evolves from the initial to the saddle state. It can be seen from figures 8 and 9 that the atoms with relatively lower $\Delta E_j$ are mainly located at or near the dislocation loop region. We observe that at some special atomic site the $\Delta E_j$ is increased, this may result from two alloying elements being too close. For example, there is a W atom at the lower atomic plane of figure 8 that shows a relatively higher $\Delta E_j$. This relatively higher $\Delta E_j$ does not change the general trend of most W atoms decreasing the $\Delta E_j$ and the increase in the value of $\Delta E_j$ is not so large because the atoms at the dislocation core have a relatively higher background $\Delta E_j$. This relatively higher 'background' $\Delta E_j$ in the dislocation core and the relatively low possibility of two alloying elements being too close means the contribution of the increase in $\Delta E_j$ by special W atoms has little effect on the overall energy decrease of $\Delta E_{act}$.

The effects of Re, W and Co on decreasing the $\Delta E_j$ of atoms mainly comes from two aspects. First, for the alloying elements in the lower and upper atomic layers across the inclined (1 1 1) slip plane, the $\Delta E_j$ for the neighbouring Ni ($L3$, $L3'$, $U6$ or $U6'$) atoms of the alloying element across the inclined (1 1 1) slip plane are decreased. The ability of Re, W and Co to lower the $\Delta E_j$ of their neighbouring Ni ($L3$, $L3'$, $U6$ or $U6'$) atoms is in the order W > Re > Co. The result here shows that the interaction between alloying elements and their neighbouring Ni atoms has an influence on the $\Delta E_j$. The Co atoms show relatively limited ability to affect the $\Delta E_j$ of their neighbouring Ni ($L3$, $L3'$, $U6$, $U6'$) atoms across the inclined (1 1 1) slip plane. This may be related to the relatively weak interaction of Co with its neighbouring Ni atoms with respect to Re or W. Second, the Re, W and Co atoms have relatively lower $\Delta E_j$ at their own atomic sites. The X (X = Re or W) atoms in the lower atomic layer across the inclined (1 1 1) slip plane have relatively lower $\Delta E_j$ especially at or near the dislocation loop region and the X (X = Re or W) atoms in the upper atomic layer across the inclined (1 1 1) slip plane does not show such an effect. The Co atoms in the upper atomic layer across the inclined (1 1 1) slip plane have relatively lower $\Delta E_j$ especially at or near the dislocation loop region, and the Co atoms in the lower atomic layer across the inclined (1 1 1) slip plane do not show this effect. This different behaviour of Co compared with Re or W needs to be further explored. The ability of Re, W and Co to lower the $\Delta E_j$ at their own atomic sites also follows the order W > Re > Co. Other randomly doped systems with the concentration of 2 at.% X (X = Re, W or Co) at the load of $K_I = 0.64K_{Ic}$ are shown in figure 10. It is shown that the above results and discussions also apply to these randomly doped systems. We can see from the results that the interaction between alloying elements and their neighbouring Ni atoms plays a role in decreasing the $\Delta E_{act}$.

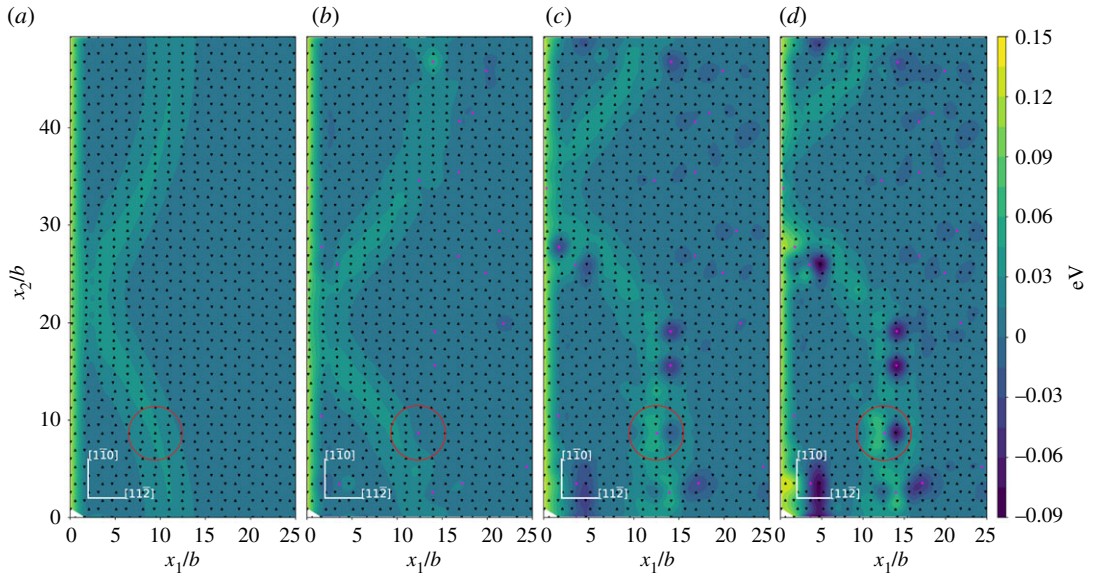

**Figure 10.** $\Delta E_j$ contour in the lower atomic layer across the inclined (1 1 1) slip plane for dislocation nucleation at the load $K_I = 0.64K_{Ic}$. (a) Ni system; (b) 2 at.%Co system; (c) 2 at.%Re system and (d) 2 at.%W system. The triangles and inverted triangles represent the upper and lower atomic layer atoms across the inclined (1 1 1) slip plane. Black triangles and inverted black triangles represent Ni atoms, pink triangles and inverted pink triangles represent X (X = Re, W or Co) atoms. Note that the alloying atom distribution is different from that in figure 8.

## 3.4. Surface energy ($\gamma_s$) and unstable stacking fault energy ($\gamma_{us}$) for different systems and their effects on dislocation nucleation at the crack tip

The static equilibrium crack can be treated as a reversible thermodynamic system according to Griffith's theory [46], in which the equilibrium between crack advance and new surface creation is established. The above equilibrium is given by the relation $G = 2\gamma_s$, where $G$ is the mechanical energy release upon crack advance and $2\gamma_s$ is the energy required for creating two new surfaces. The above relation is the necessary condition for fracture and it relates the easy cleavage plane with a relatively lower surface energy ($\gamma_s$) [47]. The unstable stacking fault energy ($\gamma_{us}$) was first proposed by Rice [11], and this quantity can be used to approximately estimate the energy barrier of partial dislocation nucleation. The ratio of $\gamma_s$ and $\gamma_{us}$ can be used to evaluate the competition between brittle and ductile behaviour [11,24,33]. A larger value of $\gamma_s/\gamma_{us}$ may result in easy dislocation emission relative to cleavage.

Molecular dynamics simulation is used for the calculation of surface energy of the $(1\,1\,\bar{1})$ plane and the calculation of unstable stacking fault energy. The model for calculating the $\gamma_s$ has the dimension $27.5a_0[\bar{1}\,\bar{1}\,2] \times 39a_0[1\,1\,\bar{1}] \times 48a_0[1\,\bar{1}\,0]$, where $a_0$ is the lattice constant for each system. The model contains 1 235 520 atoms. The surface energy is defined as $\gamma_s = (E_{surf} - E_0)/2S$, where $E_0$ is the energy of the system without free surface and $E_{surf}$ is the energy of the system after a new surface is created. $S$ is the area of the surface. $E_0$ is obtained by applying periodic boundary conditions in $[\bar{1}\,\bar{1}\,2]$, $[1\,1\,\bar{1}]$ and $[1\,\bar{1}\,0]$ directions. The lattice parameters for all the systems are also calculated based on the models for calculating $E_0$ and the calculated lattice parameters are consistent with the previous result [24]. $E_{surf}$ is obtained by applying periodic boundary conditions in the $[\bar{1}\,\bar{1}\,2]$ and $[1\,\bar{1}\,0]$ directions and free boundary condition in the $[1\,1\,\bar{1}]$ direction. The model for calculating the unstable stacking fault energy has the dimension $27.5a_0[\bar{1}\,\bar{1}\,2] \times 39a_0[1\,1\,\bar{1}] \times 48a_0[1\,\bar{1}\,0]$ and the model contains 1 235 520 atoms. The $[\bar{1}\,\bar{1}\,2]$ and $[1\,\bar{1}\,0]$ directions are periodic in space and free boundary condition is applied in the $[1\,1\,\bar{1}]$ direction. When calculating the $\gamma_{us}$, the model is divided into half crystals in the $[1\,1\,\bar{1}]$ direction and the upper half crystal is displaced gradually in the $[\bar{1}\,\bar{1}\,2]$ direction with respect to the lower half crystal.

The calculated results of $\gamma_s$, $\gamma_{us}$ and $\gamma_s/\gamma_{us}$ are listed in table 2. The Re or W addition increases the $\gamma_s$ of the Ni matrix. The higher the concentration of Re or W, the higher the $\gamma_s$. W addition has a stronger effect on increasing the value of $\gamma_s$ than Re addition at the same concentration level. The addition of Co can slightly decrease the $\gamma_s$ of the Ni matrix. When the concentration of Co changes from 1 at.% to 2 at.%, the $\gamma_s$ is lowered. With the addition of Re, W and Co, the $\gamma_{us}$ are lowered and the value of $\gamma_{us}$ decreases when the alloying element increases from 1 at.% to 2 at.%. The addition of Re, W and Co all increase the

**Table 2.** Values of surface energy ($\gamma_s$), unstable stacking fault energy ($\gamma_{us}$) and $\gamma_s/\gamma_{us}$.

| system | $\gamma_s$ (mJ m$^{-2}$) | $\gamma_{us}$ (mJ m$^{-2}$) | $\gamma_s/\gamma_{us}$ |
|---|---|---|---|
| Ni | 1499.40 (1499.3 [23]) | 281.08 (277.8 [24]) | 5.33 |
| 1 at.%Re | 1511.93 | 278.34 | 5.43 |
| 2 at.%Re | 1525.15 | 275.43 | 5.54 |
| 1 at.%W | 1520.98 | 277.35 | 5.49 |
| 2 at.%W | 1544.76 | 273.53 | 5.65 |
| 1 at.%Co | 1497.38 | 278.02 | 5.39 |
| 2 at.%Co | 1495.84 | 275.13 | 5.44 |

value of $\gamma_s/\gamma_{us}$. 2 at.% addition of Re, W or Co has a better effect on increasing the value of $\gamma_s/\gamma_{us}$ than 1 at.% addition. At the same concentration level, the ability of alloying elements in increasing the value of $\gamma_s/\gamma_{us}$ is in the order W > Re > Co. This indicates that the ability of alloying elements in promoting the dislocation nucleation at the crack tip in the Ni matrix is in the order W > Re > Co. This is consistent with the previous results.

## 3.5. Bonding between atoms calculated by discrete variational method and the implications on dislocation nucleation at the crack tip

Generally, the alloying atoms at or near the dislocation loop have a low value of $\Delta E_{act}$ as shown in the previous sections. We now examine the bonding between alloying atom and its neighbouring Ni atoms when the dislocation passes them using the *ab initio* calculation. The discrete variational method (DVM) [48,49] is used to calculate the interatomic energy (IE) [50–52] between atoms. The IE is a quantity that can be used to evaluate the bonding and interaction between neighbouring atoms, and it had been used successfully in studying the electronic structure and properties in metals and alloys [23,24,53,54]. The IE is defined as

$$E_{ll'} = \sum_n \sum_{\alpha\alpha'} N_n a_{n\alpha l}^* a_{n\alpha' l'} H_{\alpha' l' \alpha l},\tag{3.3}$$

where $N_n$ is the occupation number for the molecular orbital $\psi_n$, $a_{n\alpha l}$ given by $a_{n\alpha l} = \langle \phi_{\alpha l}(r) | \psi_n(r) \rangle$, and $H_{\alpha' l' \alpha l}$ is the interaction Hamiltonian matrix elements between atoms. A relatively larger absolute value of IE indicates a stronger interaction and bonding between neighbouring atoms [24,52,55]. More details of IE can be found in the literature [52].

We are interested in the IE of the Ni–X (X = Re, W or Co) atom pair across the inclined (1 1 1) slip plane when the dislocation passes them in the saddle state and the IE of the same Ni–X (X = Re, W or Co) atom pair across the inclined (1 1 1) slip plane in the initial state without dislocation passing by. Two kinds of Ni–X (Re, W or Co) atom pairs across the inclined (1 1 1) slip plane are considered. One kind is the Ni–X (X = Re, W or Co) atom pair where the X (X = Re, W or Co) atom resides in the lower atomic layer across the inclined (1 1 1) slip plane; the other kind is the Ni–X (X = Re, W or Co) atom pair where the X (X = Re, W or Co) atom resides in the upper atomic layer across the inclined (1 1 1) slip plane.

The alloying atoms in the centre of the red circles in figures 8 and 10 and the atoms surround them are selected as the DVM models and the models are shown in figure 11. The alloying atoms within the red circles in figure 8 are located in the lower atomic layer across the inclined (1 1 1) slip plane and the alloying atoms within the red circles in figure 10 are located in the upper atomic layer across the inclined (1 1 1) slip plane. Because the saddle state dislocation configuration may not be the same for different system models, the dislocation did not pass the Co atom within the white circle in figure 8. So, we choose the Co atom in the lower atomic layer across the inclined (1 1 1) slip plane within the red circle in figure 8 and its neighbouring atoms as the model for DVM calculation. For the Co atom in the red circle as shown in figure 8, we do not expect the general trends of the IE compared with Re and W to be vastly different. The alloying atoms all reside in the dislocation region. Furthermore, in the models shown in figure 11 the alloying atoms are not close to each other and the Co atom is in the 3d row of the periodic table, while Re and W are in the 5d row of the periodic table. The interaction of Ni–Co may not closely resemble that of Ni–X (X = Re or W) and previous calculation also showed that there is an apparent difference between the IE of Ni–Co and that of Ni–X (Re,W) [24]. The models containing the Ni–Ni atom pair

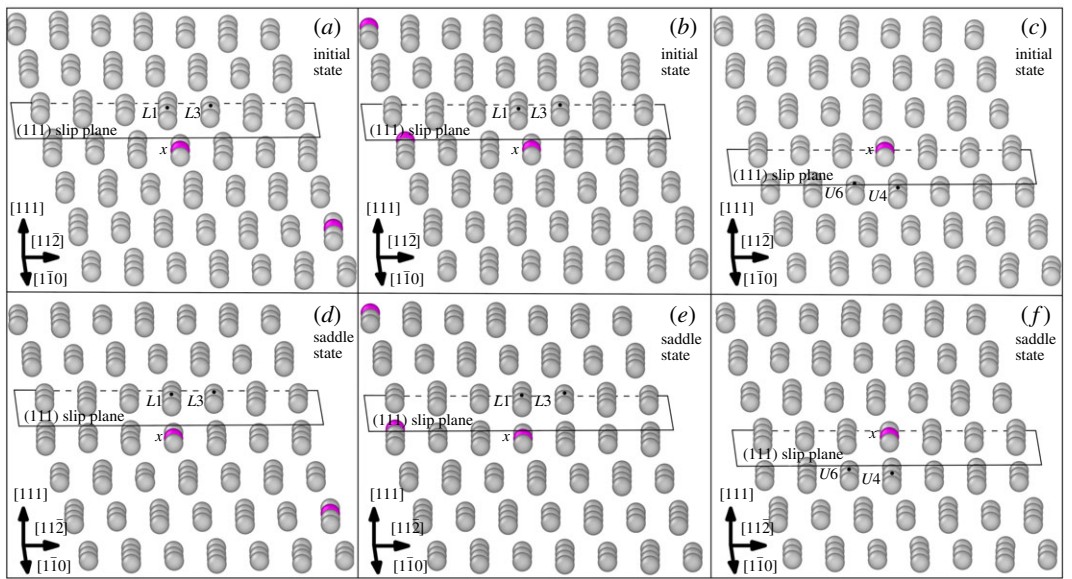

**Figure 11.** Models for DVM calculations. Grey atoms denote Ni atoms and magenta atoms denote Re, W or Co atoms. (*a*) Model with Ni−X (X = Re or W) atom pair across the inclined (1 1 1) slip plane in the initial state; (*b*) model with Ni−Co atom pair across the inclined (1 1 1) slip plane in the initial state; (*c*) model with Ni−X (X = Re, W or Co) atom pair across the inclined (1 1 1) slip plane in the initial state; (*d*) Model with Ni−X (X = Re or W) atom pair across the inclined (1 1 1) slip plane at the dislocation loop region in the saddle state; (*e*) model with Ni−Co atom pair across the inclined (1 1 1) slip plane at the dislocation loop region in the saddle state; inclined (*f*) model with Ni−X (X = Re, W or Co) atom pair across the (1 1 1) slip plane at the dislocation loop region in the saddle state.

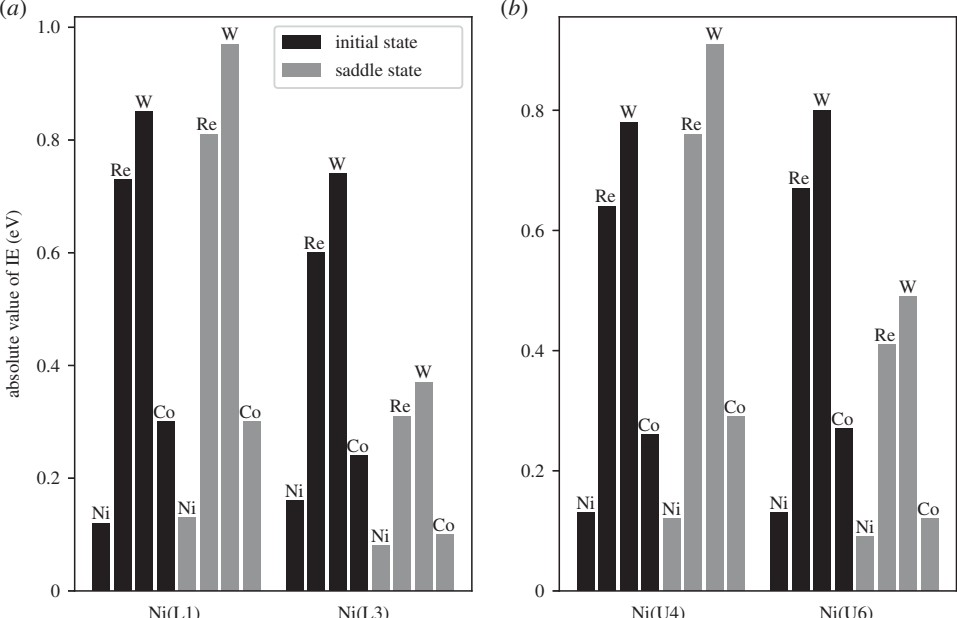

**Figure 12.** Absolute value of the interatomic energy (IE) between X (X = Re, W or Co) and its neighbouring Ni (labelled with *L*1, *L*3, *U*4 or *U*6 in figure 11) atoms across the inclined (1 1 1) slip plane in the initial state and the saddle state. (*a*) The values of IE for the models containing Ni (*L*1 or *L*3)-X (X = Re, W or Co) atom pair across the inclined (1 1 1) slip plane, where the X (X = Re, W or Co) atom of the atom pair resides in the lower atomic layer across the inclined (1 1 1) slip plane. (*b*) The values of IE for the models containing Ni (*U*4 or *U*6)-X (X = Re, W or Co) atom pair across the inclined (1 1 1) slip plane, where the X (X = Re, W or Co) atom of the atom pair resides in the upper atomic layer across the inclined (1 1 1) slip plane.

across the inclined (1 1 1) slip plane in the saddle state when dislocation passes by and the corresponding initial state models are also selected for comparison. The models each containing 171 atoms were relaxed by MD simulation with the Ni−Al−X (X = Re, W or Co) EAM potential [30−32] mentioned above. Then they are selected to perform the DVM calculations and the results for the IE are given in figure 12.

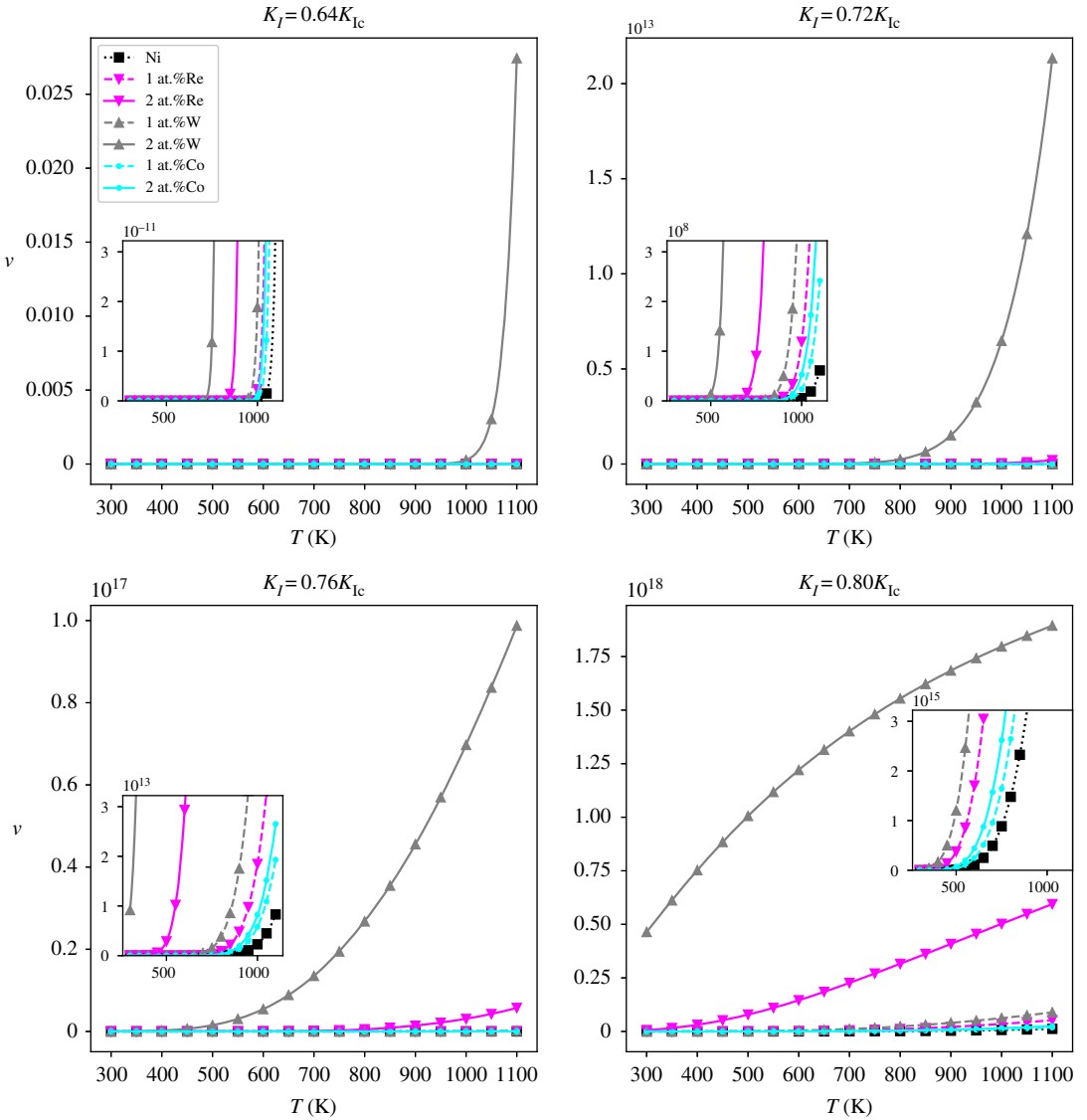

**Figure 13.** Curves of dislocation nucleation frequency ($\nu$) versus temperature ($T$) at loads $K_I = 0.64K_{Ic}$, $K_I = 0.72K_{Ic}$, $K_I = 0.76K_{Ic}$ and $K_I = 0.80K_{Ic}$.

The absolute values of IE are shown in figure 12. We can see that the absolute values of IE for Ni ($L1$, $L3$, $U4$ or $U6$)-X (X = Re, W or Co) atom pairs across the (1 1 1) slip plane are generally larger than that for the corresponding Ni–Ni atom pairs. This indicates a stronger bonding between Ni ($L1$, $L3$, $U4$ or $U6$)-X (X = Re, W or Co) atom pairs compared with the corresponding Ni–Ni atom pairs when the system evolves from the initial state to the saddle state. The absolute value of IE between the alloying element and the neighbouring Ni ($L1$, $U4$) atom across the inclined (1 1 1) slip plane becomes larger when the system evolves from the initial state to the saddle state. This can be beneficial for the dislocation emission process because the stronger interaction between the alloying element and the Ni ($L1$, $U4$) atom may make the cleavage process more difficult in terms of competition between cleavage and dislocation nucleation. The absolute value of IE between the alloying element and the neighbouring Ni ($L3$, $U6$) atom across the inclined (1 1 1) slip plane becomes smaller because the distance between these two atoms becomes larger as the model evolves from the initial state to the saddle state. We can also observe that generally W exhibits the largest absolute IE and Co the smallest. This indicates that the bonding of Ni–W is the strongest and Ni–Co the weakest among the three alloying elements studied. The stronger interaction and bonding between X (X = Re, W or Co) and Ni atoms across the inclined (1 1 1) slip plane can contribute to the stronger ability to resist the cleavage process and promote dislocation nucleation. This interatomic energy calculation validates the fact that the ability of Re, W and Co to promote the dislocation nucleation in the crack tip of Ni follows the order W > Re > Co and it is consistent with the results of previous sections.

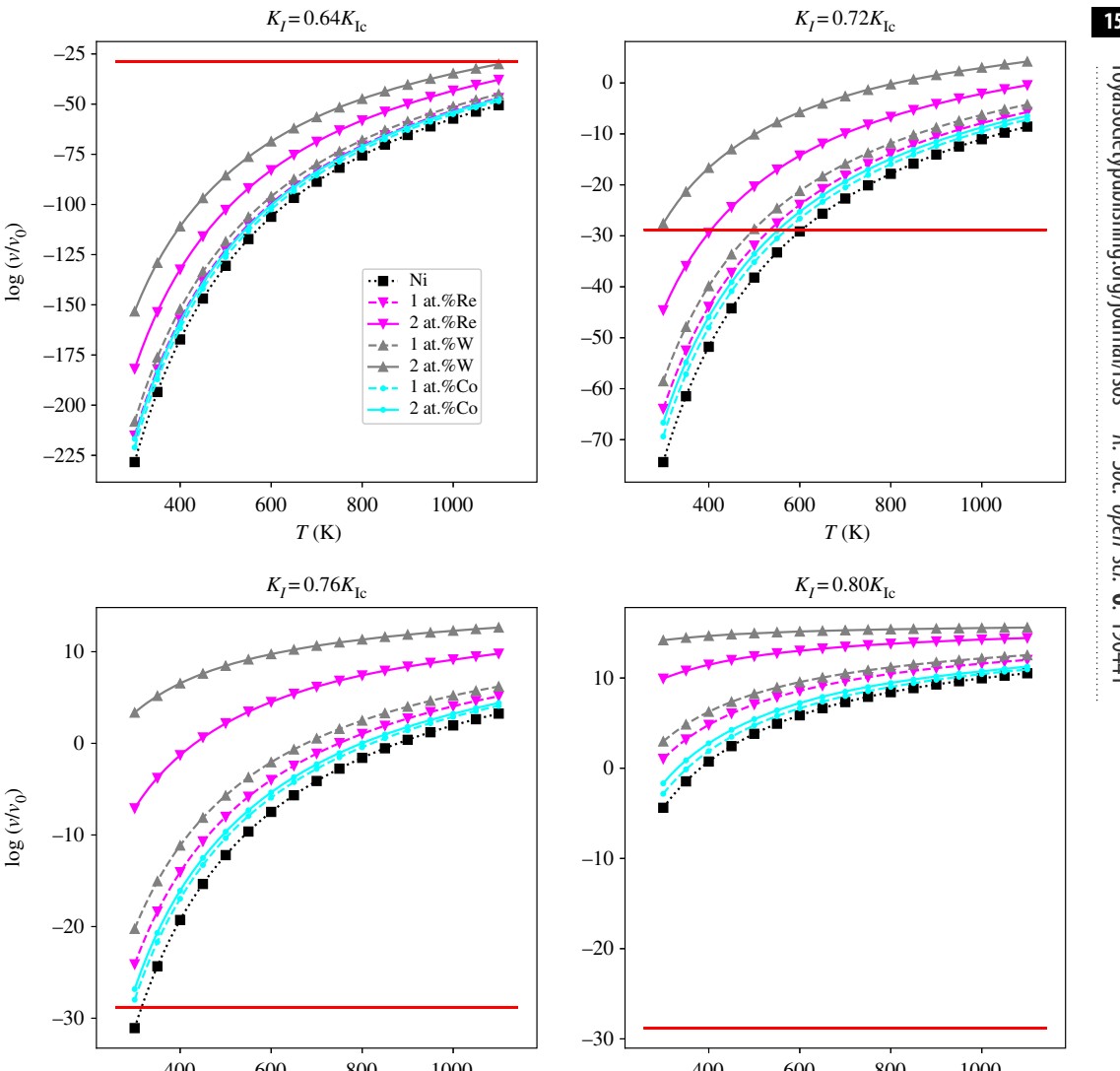

**Figure 14.** Curves of log ($\nu/\nu_0$) versus temperature ($T$) at loads $K_I = 0.64K_{Ic}$, $K_I = 0.72K_{Ic}$, $K_I = 0.76K_{Ic}$ and $K_I = 0.80K_{Ic}$. The red horizontal lines represent log ($\nu_{spon}/\nu_0$), where $\nu_{spon}$ denotes the estimated spontaneous nucleation threshold.

## 3.6. Influence of alloying element X (X = Re, W or Co) and temperature on nucleation frequency of dislocation at the crack tip

The nucleation frequency, which defines the rate of dislocation nucleation, can be estimated using the activation energy. Frequency of nucleation events per unit distance along the crack front can be estimated from [15]

$$\nu = \nu_0 \exp\left(-\frac{\Delta E_{act}}{\kappa_B T}\right) = n\left(\frac{c_{shear}}{b}\right)\exp\left(-\frac{\Delta E_{act}}{\kappa_B T}\right),\tag{3.4}$$

where $\nu_0$ is an appropriate attempt frequency and is defined as $\nu_0 = n(c_{shear}/b)$. $c_{shear} = 3$ km s$^{-1}$ is the shear wave speed. $n = 1/45b$ is the number of nucleation sites per unit length of crack front. The $45b$ is adopted in terms of the lateral spread length of the dislocation loop along the crack front. The room temperature is taken as 300K, $\kappa_B$ is Boltzmann's constant and $b$ is the Burgers vector.

When the dislocation emission occurs without a nucleation barrier, the dislocation emission becomes spontaneous. Taking $\nu_{spon} \approx 10^6 (s \cdot mm)^{-1}$ to be the thermally activated spontaneous nucleation threshold of metal in laboratory measurements [15], if the value of $\nu$ exceeds the spontaneous nucleation threshold $\nu_{spon}$, then it can be identified as spontaneous nucleation. The $\nu \sim T$ relations of different systems are shown in figure 13. To better visualize and compare the magnitude of nucleation

frequency, we plot the logarithm of the ratio of nucleation frequency and $v_0$ as a function of temperature. The log $(v/v_0) \sim T$ relation is as follows:

$$\log\left(\frac{v}{v_0}\right) = -\frac{\Delta E_{\mathrm{act}}}{\kappa_{\mathrm{B}} T}. \qquad (3.5)$$

The spontaneous dislocation nucleation threshold is thus transformed to log $(v_{\mathrm{spon}}/v_0)$. The log $(v/v_0) \sim T$ relations of different systems are shown in figure 14. The red horizontal lines in figure 14 represent log$(v_{\mathrm{spon}}/v_0)$. As the temperature increases, the nucleation frequency for each system increases. For the element types and concentrations studied in the current paper, the nucleation frequency increases as the concentration of alloying element increases. For each type of element Re, W or Co, 2 at.% concentration of alloying elements has a better effect than 1 at.% on improving the nucleation frequency of dislocation at the crack tip. And the effects of alloying elements Re, W and Co on improving the nucleation frequency of dislocation is in the order W > Re > Co when concentration changes from 1 at.% to 2 at.%. This is in line with the result of activation energy and W is the most potent element among the three elements studied in current research for increasing the nucleation frequency of dislocation nucleation. At elevated temperatures and stress intensity factors, the spontaneous nucleation of dislocation at the crack tip that can occur in terms of spontaneous nucleation frequency of dislocation is taken as $v_{\mathrm{spon}} \approx 10^6$ (s · mm)$^{-1}$.

Finally, it should be pointed out that the experimental analyses considering the alloying elements on the dislocation nucleation behaviour at the crack tip in Ni-based SC superalloys are rather scarce to our knowledge. We hope that our simulation results will stimulate related experimental efforts in this field.

# 4. Conclusion

The effects of Re, W and Co on the dislocation nucleation at the crack tip in Ni under mode I loading were studied. The results are listed below:

(i) It is found that Re, W and Co can decrease the activation energy of dislocation nucleation ($\Delta E_{\mathrm{act}}$) at the crack tip and the $\Delta E_{\mathrm{act}}$ decreases with the concentration of alloying element increasing from 1 at.% to 2 at.%. The ability of Re, W and Co to decrease the $\Delta E_{\mathrm{act}}$ is in the order W > Re > Co when concentration changes from 1 at.% to 2 at.%. The ability of Re, W and Co to promote the dislocation nucleation at the crack tip in Ni also follows the same order.

(ii) The effects of Re, W and Co on decreasing the $\Delta E_{\mathrm{act}}$ are mainly related with the decrease of the $\Delta E_j$ of their neighbouring Ni atoms and that of their own atomic sites, where $\Delta E_j$ is the energy difference of atom $j$ between the saddle state and the initial state. The ability of Re, W and Co to lower the $\Delta E_j$ follows the order W > Re > Co. This implies the interaction between alloying elements and their neighbouring Ni atoms plays a role in decreasing the $\Delta E_{\mathrm{act}}$.

(iii) It is found that the larger value of ratio $\gamma_{\mathrm{s}}/\gamma_{\mathrm{us}}$ the stronger ability of Re, W and Co to promote dislocation nucleation at the crack tip in Ni. The ability of Re, W and Co to promote the dislocation nucleation at the crack tip is in the order W > Re > Co in terms of the $\gamma_{\mathrm{s}}/\gamma_{\mathrm{us}}$ value.

(iv) Calculation and close examination of the Ni−X (X = Re, W or Co) atom pairs near the dislocation loop reveals that W generally has stronger bonding with the neighbouring Ni atoms than Re or Co does. Co has the weakest bonding effect among the three elements. The stronger bonding between the alloying element X (X = Re, W or Co) and Ni near the dislocation loop can suppress the cleavage and promote the dislocation nucleation at the crack tip.

The current study has implications for our understanding of the effects of alloying elements Re, W and Co on the ductility of the matrix phase of the Ni-based superalloys and can be helpful in the design of Ni-based superalloys.

Data accessibility. The datasets supporting this article have been uploaded as part of the electronic supplementary material.

Authors' contributions. D.W. designed the models, performed the simulation analysis and drafted the manuscript. C.W. and T.Y. provided discussions and edited the manuscript.

Competing interests. We declare we have no competing interests.

Funding. The work was supported by the National Key R&D Program of China (grant no. 2017YFB0701501 and 2017YFB0701503).

Acknowledgements. The simulations were performed on the 'Explorer100' cluster system of the Tsinghua National Laboratory for Information Science and Technology, Beijing, China.

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
