## [Reviewer comments · Royal Society Open Science]

Review History

RSOS-190441.R0 (Original submission)

Review form: Reviewer 1

Is the manuscript scientifically sound in its present form?

Yes

Are the interpretations and conclusions justified by the results?

Yes

Is the language acceptable?

Yes

Is it clear how to access all supporting data?

Yes

Do you have any ethical concerns with this paper?

No

Have you any concerns about statistical analyses in this paper?

No

Recommendation?

Accept with minor revision (please list in comments)

Comments to the Author(s)

Comments:

In this work the behavior of Re, W, and Co were studied using molecular dynamics and ab-initio calculations. The detailed analyses on energies of each atom and bonding between atoms were discussed and compared. The physics were well conducted and the conclusions were consistent in the manuscript. However, there are some doubts and misunderstandings in the manuscript. This work could be accepted as a journal in Royal Society Open Science if the authors improve the current manuscript. The questions and suggestions are as follows:

1. Upon adding Re, W, and Co into the systems, the lattice parameters will change. If the systems still use the lattice parameters of pure Ni, there will be extra stress introduced into the systems due to the different solute radius compared with that of Ni atom. The changes in their respective lattice parameters should be considered. The proper lattice parameters in the simulation would provide reliable results. Thus, whether the authors used the models with the lattice parameters of pure Ni or used the models with lattice parameters after adding alloying atoms is unclear and should be given in the manuscript.
2. There are many different crystallographic planes and cracks in the system, the reason for adopting the $(11\bar{1})[1\bar{1}0]$ crack system in the simulation should be given.
3. When the dislocation passes the W atom in the middle of Figure 8(d), the ΔE_j for this W atom is relatively higher and the ΔE_{jat} at this atomic site is not decreased, I think reasonable explanation is needed.
4. Regarding the details of the manuscript, some careful editing in the manuscript are needed:
 - (1) The unit "eV" in article should not be presented as italic font and there should be space between the numerical value and the unit. For example, in the last line of page 4, The "6.32eV" should be "6.32 eV", and here the italic font of the unit eV should also be corrected.
 - (2) In the first line of page 12, the overline of $[112]$ should not be consecutive, and should be $[\bar{1}\bar{1}\bar{2}]$ instead.
 - (3) There should be a period in the caption of each table in the main article and the supplementary material.
 - (4) There shouldn't have spaces before the expression "where equation (3.1)" in page 9.

Review form: Reviewer 2**Is the manuscript scientifically sound in its present form?**

Yes

Are the interpretations and conclusions justified by the results?

Yes

Is the language acceptable?

Yes

Is it clear how to access all supporting data?

Not Applicable

Do you have any ethical concerns with this paper?

No

Have you any concerns about statistical analyses in this paper?

No

Recommendation?

Accept with minor revision (please list in comments)

Comments to the Author(s)

The paper by Wang et al studied and compared the effect of Re, W, and Co on dislocation nucleation at the crack tip in Ni, extending their previous studies for Re in Ni. The results show that the ability of Re, W, and Co in improving the ductility of the Ni crack system is in the order $W > Re > Co$, which is consistent with their previous publications. In this sense, I think the authors should discuss how their results compare with experiments. I have other comments below.

1. Both (111) and (11-1) are slip planes in Ni. I don't see the reason why authors ignore the (11-1) plane slip which does not make sense.
2. Since the outer region of the simulation model is fixed, the author should comment on the size effect and how this fixed layer affects their results.
3. The simulations for each concentration of randomly doped alloying element are repeated three times. So it is helpful to provide the standard error for the calculated E_{act} in table 1 and Fig.3.
4. What interpolation method is used for Fig.3?
5. Why the larger ratio of surface energy to u_{sf} , the stronger ability of in promoting dislocation nucleation?
6. The results suggest that doping actually promoting dislocation. Can the author comment on how this correlates with solid solute strengthening effect?

Decision letter (RSOS-190441.R0)

28-May-2019

Dear Dr Wang:

Title: Effects of Re, W, and Co on dislocation nucleation at the crack tip in the gamma phase of Ni-based single crystal superalloys by atomistic simulation

Manuscript ID: RSOS-190441

Thank you for submitting the above manuscript to Royal Society Open Science. On behalf of the Editors and the Royal Society of Chemistry, I am pleased to inform you that your manuscript will be accepted for publication in Royal Society Open Science subject to minor revision in accordance with the referee suggestions. Please find the reviewers' comments at the end of this email. I apologise that this took longer than usual.

The reviewers and handling editors have recommended publication, but also suggest some minor revisions to your manuscript. Therefore, I invite you to respond to the comments and revise your manuscript.

Because the schedule for publication is very tight, it is a condition of publication that you submit the revised version of your manuscript before 06-Jun-2019. Please note that the revision deadline will expire at 00.00am on this date. If you do not think you will be able to meet this date please let me know immediately.

Best wishes,
Dr Laura Smith
Publishing Editor, Journals

Royal Society of Chemistry
Thomas Graham House

Science Park, Milton Road
 Cambridge, CB4 0WF
 Royal Society Open Science - Chemistry Editorial Office

On behalf of the Subject Editor Professor Anthony Stace and the Associate Editor Professor Kim Jelfs.

RSC Associate Editor:
 Comments to the Author:
 (There are no comments.)

RSC Subject Editor:
 Comments to the Author:
 (There are no comments.)

Reviewer comments to Author:
 Reviewer: 1

Comments to the Author(s)
 Comments:

In this work the behavior of Re, W, and Co were studied using molecular dynamics and ab-initio calculations. The detailed analyses on energies of each atom and bonding between atoms were discussed and compared. The physics were well conducted and the conclusions were consistent in the manuscript. However, there are some doubts and misunderstandings in the manuscript. This work could be accepted as a journal in Royal Society Open Science if the authors improve the current manuscript. The questions and suggestions are as follows:

1. Upon adding Re, W, and Co into the systems, the lattice parameters will change. If the systems still use the lattice parameters of pure Ni, there will be extra stress introduced into the systems due to the different solute radius compared with that of Ni atom. The changes in their respective lattice parameters should be considered. The proper lattice parameters in the simulation would provide reliable results. Thus, whether the authors used the models with the lattice parameters of pure Ni or used the models with lattice parameters after adding alloying atoms is unclear and should be given in the manuscript.
2. There are many different crystallographic planes and cracks in the system, the reason for adopting the $(11\bar{1})[1\bar{1}0]$ crack system in the simulation should be given.
3. When the dislocation passes the W atom in the middle of Figure 8(d), the ΔE_j for this W atom is relatively higher and the ΔE_j at this atomic site is not decreased, I think reasonable explanation is needed.
4. Regarding the details of the manuscript, some careful editing in the manuscript are needed:
 - (1) The unit "eV" in article should not be presented as italic font and there should be space between the numerical value and the unit. For example, in the last line of page 4, The "6.32eV" should be "6.32 eV", and here the italic font of the unit eV should also be corrected.
 - (2) In the first line of page 12, the overline of $[112]$ should not be consecutive, and should be $[\bar{1}1\bar{2}]$ instead.

(3) There should be a period in the caption of each table in the main article and the supplementary material.

(4) There shouldn't have spaces before the expression "where equation (3.1)" in page 9.

Reviewer: 2

Comments to the Author(s)

The paper by Wang et al studied and compared the effect of Re, W, and Co on dislocation nucleation at the crack tip in Ni, extending their previous studies for Re in Ni. The results show that the ability of Re, W, and Co in improving the ductility of the Ni crack system is in the order $W > Re > Co$, which is consistent with their previous publications. In this sense, I think the authors should discuss how their results compare with experiments. I have other comments below.

1. Both (111) and (11-1) are slip planes in Ni. I don't see the reason why authors ignore the (11-1) plane slip which does not make sense.
2. Since the outer region of the simulation model is fixed, the author should comment on the size effect and how this fixed layer affects their results.
3. The simulations for each concentration of randomly doped alloying element are repeated three times. So it is helpful to provide the standard error for the calculated σ_{act} in table 1 and Fig.3.
4. What interpolation method is used for Fig.3?
5. Why the larger ratio of surface energy to σ_{act} , the stronger ability of in promoting dislocation nucleation?
6. The results suggest that doping actually promoting dislocation. Can the author comment on how this correlates with solid solute strengthening effect?

Author's Response to Decision Letter for (RSOS-190441.R0)

See Appendix A.

RSOS-190441.R1 (Revision)

Review form: Reviewer 1

Is the manuscript scientifically sound in its present form?

Yes

Are the interpretations and conclusions justified by the results?

Yes

Is the language acceptable?

Yes

Do you have any ethical concerns with this paper?

No

Recommendation?

Accept as is

Comments to the Author(s)

I think that the revised manuscript is well completed, and should be published in Royal Society Open Science.

Decision letter (RSOS-190441.R1)

11-Jun-2019

Dear Dr Wang:

Title: Effects of Re, W, and Co on dislocation nucleation at the crack tip in the gamma phase of Ni-based single crystal superalloys by atomistic simulation
Manuscript ID: RSOS-190441.R1

It is a pleasure to accept your manuscript in its current form for publication in Royal Society Open Science. The chemistry content of Royal Society Open Science is published in collaboration with the Royal Society of Chemistry.

On behalf of the Subject Editor Professor Anthony Stace and the Associate Editor Professor Kim Jelfs.

RSC Associate Editor:
Comments to the Author:
(There are no comments.)

RSC Subject Editor:
Comments to the Author:
(There are no comments.)

Reviewer(s)' Comments to Author:
Reviewer: 1

Comments to the Author(s)

I think that the revised manuscript is well completed, and should be published in Royal Society Open Science.

Appendix A

Response to Referees ⟨ID: RSOS-190441⟩

Dear Editor(s) and Reviewers:

Thank you for your kind letter and the reviewers' comments concerning our manuscript entitled "Effects of Re, W, and Co on dislocation nucleation at the crack tip in the gamma phase of Ni-based single crystal superalloys by atomistic simulation". (ID: RSOS-190441). The comments are all very helpful for revising and improving our paper. We have studied the comments carefully and have given a detailed explanation which we hope will meet with approval. The responds to the reviewers' comments and the main corrections (the modifications are highlighted in the revised manuscript) are as follows:

Comments by Reviewer #1

1. Upon adding Re, W, and Co into the systems, the lattice parameters will change. If the systems still use the lattice parameters of pure Ni, there will be extra stress introduced into the systems due to the different solute radius compared with that of Ni atom. The changes in their respective lattice parameters should be considered. The proper lattice parameters in the simulation would provide reliable results. Thus, whether the authors used the models with the lattice parameters of pure Ni or used the models with lattice parameters after adding alloying atoms is unclear and should be given in the manuscript.

Response to Reviewer#1 comment No.1:

After doping with alloying elements, the lattice parameter of the lattice will change. When building the models, the different lattice constants for the systems with 1 at. % or 2 at. % X (X=Re, W, or Co) are considered. The calculated lattice parameters are 3.520, 3.524, 3.527, 3.525, 3.530, 3.519, 3.519 for system without alloying elements and systems with 1at.%Re, 2at.%Re, 1at.%W, 2at.%W, 1at.%Co, and 2at.%Co, respectively. This is consistent with the result of the previous calculation by Liu et al. [Shu-Lan Liu et al. RSC Adv. 2015]. According to the Reviewer's suggestion, the corresponding description is added in paragraph 3 of section 2 of the revised manuscript: "The crack systems with alloying...introduced in the following section." and in second paragraph of section 3-(d) of the revised manuscript: "The lattice parameters for...with the previous result."

2. There are many different crystallographic planes and cracks in the system, the reason for adopting the (11-1)[1-10] crack system in the simulation should be given.

Response to Reviewer#1 comment No.2:

The crack system (11 $\bar{1}$)[1 $\bar{1}$ 0] is chosen because it is experimentally observed that the cracks propagation and dislocation nucleation happen easily for this crack system. According to the Reviewer's suggestion the related description and supportive references are added in the revised manuscript. Please see the second paragraph of section 2 in the revised manuscript: "This crack system is chosen...for this crack system".

3. When the dislocation passes the W atom in the middle of Figure 8(d), the ΔE_j for this W atom is relatively higher and the ΔE_j at this atomic site is not decreased, I think reasonable explanation is needed.

Response to Reviewer#1 comment No.3:

This may be resulted from two alloying elements being too close. The relatively higher ΔE_j does not change the general trend of most alloying W atoms decrease the ΔE_j and the increase in the value of ΔE_j is not so large because the atoms at the dislocation core have a relatively higher background ΔE_j . This relatively higher "background" ΔE_j in the dislocation core and

the relatively low possibility of two alloying elements being too close (for the concentration in current study) makes the contribution of the increase in ΔE_j by special W atoms have little effect on the overall energy decrease of ΔE_{act} . According to the Reviewer's suggestion, we added the corresponding explanation in the second paragraph of section 3-(c) in the revised manuscript: "We observe that at some...overall energy decrease of ΔE_{act} ".

4. Regarding the details of the manuscript, some careful editing in the manuscript are needed:

(1) The unit "eV" in article should not be presented as italic font and there should be space between the numerical value and the unit. For example, in the last line of page 4, The "6.32eV" should be "6.32 eV", and here the italic font of the unit eV should also be corrected.

(2) In the first line of page 12, the overline of [112] should not be consecutive, and should be [-1 -1 2] instead.

(3) There should be a period in the caption of each table in the main article and the supplementary material.

(4) There shouldn't have spaces before the expression "where equation (3.1)" in page 9.

Response to Reviewer#1 comment No.4 (1):

We are very sorry for our carelessness in expressions and symbols. According to the Reviewer's suggestion we checked the whole manuscript and make sure there was a space between the numerical value and the unit. To avoid similar errors, we checked all the symbols in the manuscript and corrected the similar expressions.

Response to Reviewer#1 comment No.4 (2):

We consulted the relevant books and references and found that the crystallography directions are indeed written in this form. We checked the whole manuscript and corrected the relating expressions according to the Reviewer's suggestion.

Response to Reviewer#1 comment No.4 (3):

According to the Reviewer's suggestion, we added periods at each caption of the tables in the revised manuscript and the revised supplementary material.

To avoid similar errors, we also checked all the captions of the figures in the manuscript and added periods at each caption of the figures in the revised manuscript.

Response to Reviewer#1 comment No.4 (4):

According to the Reviewer's suggestion, the spaces before the expression "where equation (3.1)" in page 9 is removed. We also checked other places in the manuscript to avoid similar errors.

Comments by Reviewer #2

The paper by Wang et al. studied and compared the effect of Re, W, and Co on dislocation nucleation at the crack tip in Ni, extending their previous studies for Re in Ni. The results show that the ability of Re, W, and Co in improving the ductility of the Ni crack system is in the order $W > Re > Co$, which is consistent with their previous publications. In this sense, I think the authors should discuss how their results compare with experiments.

Response to Reviewer#2 comment No.0:

Considering the comparison of our simulation results with the experiments. P. Li et al. [P. Li et al. Int. J. Fatigue, 2014] studied the low-cycle-fatigue behaviours of the Ni-based single crystal superalloys without (0 wt.%) and with Re addition (3 wt.%), however the behaviour of dislocation nucleation from the crack tip is not addressed. Y. B. Xu et al. [Y. B. Xu, R. J. Wang, and Z. G. Wang, Phys. Stat. Sol. (a), 1994][Z. G. Wang and Y. B. Xu, Defect and Diffusion Forum, 1997] studied the Ni-based single crystal superalloy DD8 during deformation by in-situ experiments. Although they observed the evidence of dislocation emission from the crack tip, the crack deformation mode for the dislocation emission in their study is mode III and does not directly support our simulation results. Moreover, their study also did not elucidate the alloying effects on the dislocation nucleation behaviour at the crack tip. Until now, the direct experimental result of alloying element effects on the dislocation nucleation at the crack tip is scarce to our knowledge. Due to the importance of failure and fracture mechanisms in the materials and the fact that the direct experimental analyses of alloying effects on dislocation nucleation from the crack tip related in our study is still lacking, we hope our simulation results will stimulate the experimental efforts in this field. According to the Reviewer's suggestion, the related description is added in the last paragraph of section 3: "Finally, it should be pointed out ... experimental efforts in this field."

1. Both (111) and (11-1) are slip planes in Ni. I don't see the reason why authors ignore the (11-1) plane slip which does not make sense.

Response to Reviewer#2 comment No.1:

This issue can be elucidated from two aspects.

First, the dislocation nucleation will happen in a plane which subjects to high shear stress. The $(11\bar{1})$ plane is coplaner with the crack plane in our study. Under mode I loading, the slip plane with the highest shear stress happens at the plane that forms an angle $\theta=70.53^\circ$ with the crack plane, i.e. the dislocation nucleation plane is not coplaner ($\theta=0^\circ$) with the crack plane [James R. Rice, J. Mech. Phys. Solids, 1992]. Thus, if the crack plane lies in $(11\bar{1})$ plane, the dislocation nucleation plane will be in the (111) plane.

Second, the $\{111\}$ slip planes in FCC structure includes 4 sets of octahedral planes, each set of octahedral planes includes two parallel planes. The 4 sets of octahedral planes are (111) , $(11\bar{1})$, $(\bar{1}11)$ and $(1\bar{1}1)$ planes. When the crack plane lies in either of these 4 planes, the dislocation nucleation and slip will happen on the plane that forms an angle $\theta=70.53^\circ$ with the crack plane under mode I loading.

From the above two aspects, the model with the dislocation nucleating at (111) plane from the $(11\bar{1})$ crack plane is reasonable.

2. Since the outer region of the simulation model is fixed, the author should comment on the size effect and how this fixed layer affects their results.

Response to Reviewer#2 comment No.2:

Our study focused on the comparative study of the effects of Re, W, and Co on dislocation nucleation behavior at the crack tip, which mainly discussed the configurations and energetics relating to the saddle state during the dislocation nucleation process. As can be seen from Fig.5 in the main article, the dislocation loop near the crack tip is localized in the saddle state. Thus, the fixed boundary effect on our transition state should be weak and the in-plane radius R in our model is reasonable from this respect. The fixed boundary condition should have strong influence on the final state configuration (the dislocation moves far away from the crack tip) and energetics, which is not our concern in current study. Regarding the model size along the crack front length, the activation energy of dislocation nucleation grows almost linearly with the crack front length in the pure Ni system under the condition $K_I = 0.64K_{Ic}$, which is shown in Figure 1 in the revised supplementary material. This result is reasonable as the saddle state dislocation spans the whole crack front

length and the dislocation nucleation is expected to be more difficult with the increase of the dislocation line length. Thus, the current model size is reasonable and it also makes the simulations computationally feasible in our study. According to the Reviewer's suggestion, the related comments on the size and fixed layer effects are added in the second paragraph of section 3-(b) in the revised manuscript: "It can also be seen that the dislocation loop ... in our model is reasonable from this respect." and in the revised supplementary material: "The test of the model size ... activation energy of an isolated dislocation.". The Figure 1 in the revised supplementary material is also added as the supporting material.

3. The simulations for each concentration of randomly doped alloying element are repeated three times. So it is helpful to provide the standard error for the calculated Eact in table 1 and Fig.3.

Response to Reviewer#2 comment No.3:

According to the Reviewer's suggestion, the error bar of the standard error is added into Fig.3 of the revised manuscript and the values of the standard errors are added into Table 1 of the revised manuscript.

4. What interpolation method is used for Fig.3?

Response to Reviewer#2 comment No.4:

We are very sorry for the omission of mentioning the interpolation method in Fig.3. The interpolation method in Fig.3 we use is the cubic-spline interpolation. According to the Reviewer's suggestion, the related description is added in the second paragraph of section 3-(a) in the revised manuscript: ", in which the cubic-spline interpolation is used."

5. Why the larger ratio of surface energy to σ_{sf} , the stronger ability of in promoting dislocation nucleation?

Response to Reviewer#2 comment No.5:

The factors that can affect the toughness and/or ductile include dislocation mobility and crack tip blunting [James R. Rice and Robb Thomson, Philos. Mag., 1974]. The description of the brittle vs. ductile behaviour should not

only consider the effect of cleavage decohesion, but also should include the dislocation emission process. In terms of the Rice-Thomson approach [James R. Rice and Robb Thomson, Philos. Mag. 1974, James R. Rice, J Mech Phys Solids, 1992], the brittle vs. ductile behaviour is based on a comparative analysis of two competing process: (1) The opening of the crack, and (2) The dislocation emission at the crack tip.

The resistance to dislocation emission at the crack tip can be measured with the maximum energy related with the sliding of the atomic planes (γ_{us}) [James R. Rice, J. Mech. Phys. Solids, 1992]. The larger the γ_{us} is, the more difficult the dislocation nucleation at the crack tip.

The cleavage decohesion is related with the magnitude of the surface energy. The larger the surface energy, the more difficult the cleavage decohesion will be and the more difficult two new surfaces will form. Either the dislocation nucleation at the crack tip or cleavage decohesion can release the induced stress and strain in the system. As mentioned above, the ductile-brittle response is neither solely the result of dislocation emission nor the solely result of the cleavage decohesion, but the result of the competition between the two processes. Thus, the comparative analysis of the two competing processes can be combined (by adopting the ratio of γ_s and γ_{us}) to characterize the ease of the dislocation nucleation.

6. The results suggest that doping actually promoting dislocation. Can the author comment on how this correlates with solid solute strengthening effect?

Response to Reviewer#2 comment No.6:

The possible mechanisms for solid solution strengthening are: (1) The ability of the alloying effects in decreasing the diffusivity and the stacking fault energy [Ernst Fleischmann et al. Acta Mater. 2015]. (2). The combined effect of the solid solution strengthener in partitioning in the γ phase and the strong interaction of solid solution strengthener with the dislocation in the γ phase and retard the dislocation motion [M. Pröbstle et al. Mater. Sci. Eng. A, 2016][Feng-Hua Liu and Chong-Yu Wang, RSC Adv. 2017]. (3) Other mechanisms may also include the Cottrell atmosphere of alloying elements in pinning the dislocations etc.

The solid solution strengthener may induce the strengthening effect to the alloys by the above possible mechanisms, which mainly lies in the effects

of alloying elements in decreasing the diffusivity, stacking fault energy and retarding dislocation motions etc.

From the microscopic view, the two categories (solid solution strengthening and the dislocation nucleation at the crack tip) have different mechanisms in influencing the mechanical properties of the material. From the macroscopic view, they have some connections. The “strengthening” of the material does not necessarily mean the material is more “brittle”.

The final rupture of the material is usually displayed with the cracks. The ductile vs. brittle behaviour of the material can be explained by the competition between the dislocation nucleation and cleavage in the alloys. The alloying elements Re, W, and Co can promote the dislocation nucleation against the cleavage process, which can suppress the extension and propagation of the crack. Thus, the ability of resisting the cleavage failure is improved. It is desired for material to have reasonable ductility to prevent the components and structures from catastrophic failure during service. This is not contradictory to the solute solution strengthening effect, as the resistance to cleavage failure and the solute solution strengthening can both be beneficial to the material.

Other modifications made to the revised manuscript:

1. In our revised manuscript, the use of more factual data and arguments are added to enhance the article rigor and persuasiveness. To this end, we give another simulation result with a different doping. The results and conclusions of the article are not affected by the factual data added. The data is added as Table 4 in the revised supplementary material and the corresponding arguments and descriptions are added in the second paragraph of section 3-(a) in the revised manuscript: "...are repeated four times" and "(We also observe the phenomena ... random fluctuation of atoms.)". The Table1, Figure 3, Figure 13, and Figure 14 are updated at the same time. The results are reasonable and consistent with our conclusion.
2. All the text "Fig." in the article are modified to "Figure" in the revised manuscript.
3. We added spaces between the texts and the left parentheses "(" in the revised manuscript. For example, "X(X=Re, W, or Co)" is modified to " X (X=Re, W, or Co)".
4. An author Tao Yu has been added in our article, who contributed to the discussion, editing the manuscript and improvements of the article. The content of the section "Authors' Contributions" is modified as: "Chongyu Wang and Tao Yu provided discussions and edited the manuscript."
5. The email address for the author Chongyu Wang is corrected from cywang@mails.tsinghua.edu.cn to cywang@mail.tsinghua.edu.cn.
6. The grant number is modified as "The work was supported by the National Key R&D Program of China (Grants No.2017YFB0701501)."
7. In the "References" section, we added references which is labeled with the reference numbers of 34, 35, and 36.
8. In the supplementary material, the Figure 1, Table 4 and the corresponding description are added.
9. Other grammatical and spelling mistakes (including those pointed out by the Reviewers) are corrected and highlighted (red colored texts) in the main article.